



# Aeolus Lidar Surface Returns (LSR) at 355 nm as a new Aeolus L2A Phase-F product

Lev D. Labzovskii[1], Gerd-Jan van Zadelhoff[1], David P. Donovan[1], Jos de Kloe[1], L. Gijsbert Tilstra[1], Ad Stoffelen[1], Damien Josset[2] , Piet Stammes[1]

[1]R&D Satellite and Observations Group (RDSW), Royal Netherlands Meteorological Institute (KNMI), De Bilt, the Netherlands
[2]U.S. Naval Research Laboratory, NASA Stennis Space Center, MS 39529, USA

*Correspondence to*: lev.labzovskii@knmi.nl

**Abstract:** The Atmospheric Laser Doppler Instrument (ALADIN) onboard Aeolus was the first spaceborne high-resolution

lidar, measuring vertical profiles of aerosol optical properties at 355 nm at an incidence angle of ~35°. Although Aeolus had been primarily developed to provide vertical profiles of wind speed, aerosols, and cloud products, its lidar surface returns (LSR) were shown to contain useful information about UV surface reflectivity and agreed well with passive remote sensing reflectance. Within the process to incorporate the LSR algorithm into the Aeolus Level 2A product during the post-commissioning phase of Aeolus, we describe the methodology and evaluate the results of the adopted LSR retrieval. The

algorithm combines attenuated backscattering parameters (L2 AEL-PRO data) with the information on the surface bin detection (L1 data) to produce attenuated LSR estimates (e.g. surface integrated attenuated backscatter) for all bins where the ground was detected. The correction for producing final LSR estimates at the original Aeolus resolution is performed using the Aeolus L2 retrievals, namely Aerosol Optical Depth (AOD) and Rayleigh Optical Depth to ensure that LSR is free from effects of atmospheric attenuative features such as optically thick clouds and thick aerosol conditions (AOD > 1.0). The

evaluation shows that Aeolus LSR estimates produced from this approach agreed well with the UV Lambertian-Equivalent Reflectivity (LER) from GOME-2 (LER$_G$) and TROPOMI (LER$_T$) climatologies at all spatial scales. For four reference orbits (September 10, 2018; November 30, 2018; January 11, 2019; and May 1, 2019), all cloud and aerosol-free LSR estimates agree well with both LER references with correlation coefficients (r) varying from 0.55 to 0.71. For monthly scales, the agreement was moderate-to-high for LSR-LER$_T$ (r = 0.61 – 0.77 depending on the month) and was weak-to-

moderate for LSR-LER$_G$ comparison (r = 0.44 – 0.64). Globally, the averaged 2.5 x 2.5º LSR estimates exhibit very high agreement with both LERG (0.90) and LERT (0.92) references. In reproducing regional monthly dynamics LSR and LER agree very well in snow/ice-covered regions (r > 0.90), semi-arid regions (r > 0.90), arid regions (r > 0.70), and only some regions with mixed vegetation like Australia (r = 0.94), while no agreement was found for ocean regions due to the Aeolus optical setup, favourable for ocean subsurface, not direct surface backscatter probing. We unveiled four reflectivity clusters

of LSR at 2.5 x 2.5 degree grids, manifesting a transition from white to darker surfaces in descending LSR magnitude order: ice, snow, surface without snow, and water. Regionally, the LSR-LER agreement can vary and yields the highest correlation values in regions where snow is present in winter. This pattern is explained by the very good sensitivity of LSR to modelled





snow cover we demonstrated (r = 0.62 – 0.74 between these parameters in such regions), while sensitivity to purely vegetation-driven changes of surface is lower, as indicated by the comparison between LSR and NDVI without snow (r <

0.30 in the regional analysis). Overall, our work complemented existing LSR studies that were mostly focused on nadir-looking CALIPSO cases by demonstrating the usability of LSR for scientific applications at non-nadir angles. By taking together CALIPSO and Aeolus experiences, a framework on effective LSR utilization using future lidar missions such as EarthCARE and Aeolus-2 can be effectively designed.

# 1 Introduction

Most spaceborne nadir-looking lidars have been and are being developed for scientific applications focused on the atmosphere given the ability of lidars to provide multi-year vertical profiles of aerosols and clouds [Winker et al., 2010] in the case of CALIPSO and wind distribution additionally in the case of Aeolus [Lux et al., 2010]. Besides these main applications, spaceborne lidars can register backscattered echoes that are formed after a lidar beam interacts with the land or water surface. Earlier lidar studies had suggestively demonstrated that backscatter reflectance coming from surfaces may

strongly vary depending on the reflector in both laboratory [Kavaya, 1983] and field campaign conditions [Reagan and Zelinskie, 1991]. Although most earlier studies had been focused on the surface as a target for lidar calibration [Kavaya, 1983; Cooley et al., 1993], later works have also turned their scientific interest toward lidar surface returns (hereafter – LSR) as a proxy for deriving physical characteristics of land/ocean surface or atmosphere [Josset et al., 2018]. Such opportunity emerged due to the significantly stronger surface backscatter signals compared to atmospheric signal registered at lidar

detector [Venkata and Reagan, 2016]. In this research domain, studies which used CALIPSO observations extensively exploited ocean returns and repeatedly demonstrated that for visible and infrared wavelengths at nadir incidence, LSR can be most effectively used to infer ocean surface [Josset et al., 2008] and subsurface [Lu et al., 2014] conditions. These findings further prompted a number of successful, mostly CALIPSO application, studies (but not only, see Dimitrovic et al., [2023]), focused on either derivation of atmospheric characteristics (wind speed or aerosol) using ocean surface return [Hu et al.,

2008; Josset et al., 2008; 2010; He et al. 2016] or oceanic organic carbon chlorophyll using ocean subsurface return [Lu et al., 2014; 2021; Behrenfield et al., 2016] parametrizations. Besides that, there were initiatives to exploit CALIPSO surface reflectance signal from land (for instance, to use two identical reflectance tracks) for further inferring atmospheric characteristics such as aerosol optical depth [Josset et al., 2018].

Most recent lidar surface return-focused studies were conducted for the recently decommissioned missions of

CALIPSO and Aeolus. Overall, despite their considerable instrumental differences, both CALIPSO [Lu et al., 2018] and Aeolus [Labzovskii et al., 2023] were shown to be sensitive to surface reflectivity characteristics. These recent findings opened an interesting research niche. It has become increasingly clear that spaceborne lidars can substantially complement our knowledge on the surface reflectivity which had been previously based on the information from passive remote sensing instruments only, limited by daylight observations and suffering from large low solar angle-driven uncertainties at high



latitudes [Tilstra et al., 2017; 2024]. Despite these promising developments, our current knowledge about LSR is still mostly based on numerous CALIPSO studies, while Aeolus works on LSR remained scanty. This knowledge gap has emerged due to the challenges to retrieve robust LSR from Aeolus given its unique instrumental setup (incidence angle of ~35° and 355-nm wavelength). The main challenges we refer to include very weak backscatter signals from the water surface, coarse lidar surface bins of Aeolus [Ehlers et al., 2022] potentially containing subsurface, surface and atmospheric components, weak sensitivity and a lower sensitivity of Aeolus surface backscattering driven by the unique Aeolus setup. Most of these LSR-related challenges were briefly reported by preliminary exploratory Aeolus studies [Dionisi et al., 2023; Jamet et al., 2023; Labzovskii et al., 2022] or were touched upon in Aeolus studies pointedly focused on atmospheric retrievals [Weiler et al., 2021; Gikas et al., 2023]. The potential roadblocks for studying Aeolus LSR had been long anticipated from both theoretical considerations [Josset et al., 2010b] and from Aeolus pre-launch preparations, based on the Aeolus airborne demonstrator, designed to be identical to the spaceborne ALADIN [Li et al. 2010; Weiler, 2017]. Despite the physical constraints of Aeolus ocean surface returns, our previous study has shown that Aeolus land LSR agreed with previous estimates of passive remote sensing surface reflectivity with some differences in the way how reflectivity varies across different land types [Labzovskii et al., 2023]. At the same time, our previous study has not described or documented the methodological approach to derive useful Aeolus LSR estimates, was limited to seasonal dynamics of one region (the Sahara), while showing only comparison of LSR versus LER references at gridded (2.5 x 2.5°) and regional (regional averages were compared) resolutions.

In this light, this paper has two main objectives including (a) presentation of the detailed Aeolus LSR retrieval methodology which will be incorporated in the official Aeolus reprocessing chain (Level 2A) data as well as (b) very detailed LSR regional analysis that continues the effort of the Labzovskii et al. [2023] letter. In line with the availability of the latest reprocessing results, this study is performed for the Flight Mode-A period of Aeolus (09.2018 – 05.2019). The rest of the paper is organized as follows. Section 2 presents Data and Methodology, while the Sections 3, 4 and 5 represent Results, Discussion and Conclusions, respectively.

## 2. Data and Methodology

This section describes the data and methodology applied in this study.

### 2.1 Data

### 2.1.1 Aeolus data

Aeolus was launched on 22 August 2018 to measure atmospheric wind profiles from the ground to the stratosphere at global scales and remained spaceborne until July 2023 [Stoffelen et al., 2005; Reitebuch et al., 2019; Lux et al., 2020]. Aeolus carried the Atmospheric Laser Doppler Instrument (ALADIN), a high-spectral resolution UV lidar (355 nm) pointing at an incidence angle of ~35°. The instrument provided information on the lowest ~30 km of the atmosphere (0.25 – 2.00 km





vertical resolution depending on the altitude) for 15.6 orbits per day with a 06:00 and 18:00 local solar time (LST) Equator overpass in a sun-synchronous orbit. Aeolus was the first lidar instrument in space to measure the Doppler shift, using a

High Spectral Resolution Lidar (HSRL) technique, from both two lidar channels (Mie and Rayleigh). Next to wind information, Aeolus provided vertical information about aerosols and clouds [Flament et al., 2021; Ehlers et al., 2022] and surface backscattering echoes [Labzovskii et al., 2023], i.e. LSR, that we examine in this study. For calculating LSR the following data are used: Aeolus L1B [Reitebuch et al., 2018] and Aeolus AEL_PRO data [Donovan et al., 2022]. For additional analysis over ocean that requires modelled wind information, we also ingest auxiliary meteorological data, from

the Aeolus (AUX_MET data) ground segment, compiled using ECMWF winds [Lux et al., 2022].

Table 1 illustrates the Aeolus data required to produce LSR. This study uses Aeolus data including the Level 1B (L1B) reprocessing product for detecting the surface at the highest spatio-temporal resolution of Aeolus sounding. The methodology relies on, but not precisely follows, the procedure described briefly in Labzovskii et al. [2023]. The complete and explicit explanation of the current approach is provided in more details further. The L1B data provide basic Aeolus

information required for calculating LSR including measurement geolocation (longitude, latitude, altitude of lidar bin, width of the range gate and most importantly – in which lidar bins the surface is located), L2 AEL-PRO provides essential data about atmospheric optical characteristics (including attenuated particle backscatter, aerosol extinction, molecular backscatter; all provided alongside their associated uncertainties and scene classification) [Donovan et al., 2023]. Note that L1B and L2A data types are both required to first produce AEL-FM (feature mask) data originally developed for the ATLID instrument of

the future EarthCARE mission [van Zadelhoff et al., 2023] for classifying atmospheric features such as clouds that can critically attenuate our LSR estimates. All L1B, L2A and AEL_FM are further used to produce AEL_PRO data which represents a cornerstone dataset for retrieving robust LSR values, as shown earlier in our previous work [Labzovskii et al., 2023]. We emphasize that AEL-PRO is using an optimal estimation [Rodgers 2000] forward-modelling inversion procedure, describe by Donovan et al. [2022, 2023] in detail.


**Table 1. Aeolus data used as input in this study**

| Type of data | Version | Purpose |
|---|---|---|
| L1B | L1bP v7.12 | Input for producing AEL_FM |
| | | Input for producing AEL_PRO |
| | | Input for detecting ground bin |
| L2A | - | Input for producing AEL_FM |
| | | Input for producing AEL_PRO |
| AEL_FM | 1.70 | Input for producing AEL_PRO |
| AEL_PRO | 1.72 | Source of information about cross-talk- |



| | | calibrated data: extinction coefficient, backscattering coefficient, Rayleigh backscattering |
|---|---|---|

Although this paper is focused on the entire FM-A period, we have selected several orbits as illustrative examples for more detailed analysis of the underlying LSR data. Table 2 below shows four reference orbits from 10 September 2018, 30 November 2018, 11 January 2019 and 01 May 2019 with the exact L1B and AEL-PRO files applied in the analysis. The orbits were selected to represent as different seasonal and geographic conditions as possible for such a narrow selection of orbits.

**Table 2. Reference orbits used as examples in the methodology**

| orbit # | L1B file | AEL-PRO file |
|---|---|---|
| 1 | AE_OPER_ALD_U_N_1B_20180910T170826021_005556004_000299 | AEL_PRO_20181130T092250030_005411999 |
| 2 | AE_OPER_ALD_U_N_1B_20181130T092250030_005411999_001578 | AEL_PRO_20181130T092250030_005411999 |
| 3 | AE_OPER_ALD_U_N_1B_20190111T010350031_008363990_002238 | AEL_PRO_20190111T010350031_008363990 |
| 4 | AE_OPER_ALD_U_N_1B_20190501T003753023_005268013_003982 | AEL_PRO_20190501T003753023_00526801 |

### 2.1.2 Data for validating Lidar Surface Returns from Aeolus: TROPOMI and GOME-2 Lambertian Equivalent Reflectance (LER)

From a validation perspective, we followed the methodology from our previous paper [Labzovskii et al., 2023] and used Lambertian-equivalent reflectivity (LER) estimates from TROPOMI (TROPOspheric Monitoring Instrument) and GOME-2 (Global Ozone Monitoring Experiment–2), referred to as LERT and LERG, respectively. Surface LER represents the reflectivity of the surface that was retrieved using the assumption of Lambertian surface reflection. In reality, most surfaces do not behave as a Lambertian reflector. We acquired TROPOMI (minimum LER with snow/ice v2.0 (accessed from https://www.temis.nl/surface/albedo/tropomi_ler.php) and GOME-2 LER (mode LER, v4.0) at 354 nm (the closest wavelength to the 355 nm wavelength of Aeolus) monthly climatologies [Tilstra et al., 2017; Tilstra et al., 2024] with the highest spatial resolution available of 0.125° × 0.125° and 0.25° × 0.25°, respectively. The TROPOMI surface DLER database comprises LER estimates for the Earth's surface across 21 one-nanometre (nm) wide wavelength bands between 328 and 2314 nm. This contains the directionally-dependent Lambertian Equivalent Reflectivity (DLER) of the surface, with an increased precision of the surface reflectance across a range of viewing angles due to the Bidirectional Reflectance Distribution Function (BRDF). This development is a considerable improvement for the reflectance, particularly within the longer wavelength bands. The latest version of LERT (2.0) is based on 60 months of TROPOMI data (02.01.00 L1B



reprocessing data) with improved cloud filtering of scenes applied [Tilstra et al., 20243 preprint]. Both LER estimates were resampled to 2.5° x 2.5 ° grids with the uncertainties as the standard deviation of LER during the month. LER was

downloaded from the TEMIS (Tropospheric Emission Monitoring Internet Service) website. We used LER estimates, reflecting snow-affected areas as well because it is crucial to include snow/ice regions in the analysis to evaluate the sensitivity of Aeolus LSR to the strongest white reflectors. Note that our previous paper demonstrated that although LER and LSR represent Lambertian and unidirectional reflectivity characteristics [Labzovskii et al., 2023]. As such, while LER is the most suitable reference for global surface UV reflectivity, perfect linear agreement with LSR is not expected.


### 2.1.3 Land-cover related reference data

To verify our previous suggestion on the  strong sensitivity of LSR to snow cover and moderate sensitivity to vegetation type

[Labzovskii et al., 2023], we used reference datasets including snow cover and NDVI (Normalized Difference Vegetation Index). Snow cover data were taken from the MERRA-2 (Modern-Era Retrospective analysis for Research and Applications version 2) model, namely from the M2TMNXGLC dataset, where the fractional area covered by snow on glaciated surfaces ranges from 0 to 1. NDVI is a measure used to gauge the health and density of vegetation on land surfaces. It is calculated as follows: NDVI = (NIR - VIS)/(NIR + VIS), where NIR is near-infrared radiation and VIS is visible wavelength radiation.

NDVI ranges from 0 to 1, where higher NDVI values indicate healthier vegetation and lower values suggest sparse or stressed vegetation, bare soil, or non-vegetated areas. NDVI data also originate from the MERRA-2 records (M2TMNXLND). Note that both snow cover and NDVI data were taken from MERRA-2 version 5.12.4 [GMAO, 2015]. They were accessed and downloaded using the  GEOVANNI tool of the NASA EarthDATA portal (accessed on 25.02.2024 last time).


### 2.2 Methodology

### 2.2.1 Ground bin detection and calculation of LSR

ALADIN was a unique spaceborne lidar instrument with the range gate setting varying depending on the location [Reitebuch et al., 2018]. One cannot simply select a fixed lidar bin number, corresponding to the surface intersection based on some

orbit example. Due to this, our first Aeolus LSR-focused work [Labzovskii et al., 2023] followed the experience of pre-Aeolus lidar studies [Josset et al., 2018] and sought for the minimum difference between the altitude where DEM is located according to the model implemented in Aeolus data and the respective Aeolus range gate (among 24 Aeolus bins). The current paper aligns the ground bin detection for LSR retrieval with the Aeolus official processing approach and takes the information where ground is located from L1 data, namely, from ground wind detection block [Lux et al., 2018]. Hereafter,



we refer to this approach as "official" for brevity. The official approach differs from Labzovskii et al. [2023] method of ground bin detection (Labzo-23 for brevity) as follows. The assumptions of Labzo-23 approach were: (1) ground signal is present only in the Aeolus bin, closest to DEM and (2) that at clear atmospheric conditions, ground bin can be always detected regardless the signal strength. The assumptions of the currently applied, official approach are different: (1) the surface signal can be distributed across several Aeolus bins and (2) sometimes, ground signal cannot be detected (signal is

too weak), which means that some Aeolus observations do not contain any surface signal even over land. To understand the mechanism behind the official Aeolus ground detection algorithm, note that it uses the GTOPO30 global model containing DEM ACE v.1 information at high resolution (300 m x 300 m, 9 arcsecond resolution). The height of the surface of the Earth with regard to the reference ellipsoid is used. Subsequently, the lower edge of each altitude bin is being sought, where the height of the bin should be below height of DEM. The algorithm is searching for the first bin starting from altitude with non-

negative valid useful signal (uppermost ground bin candidate). Subsequently, the algorithm seeks for a significant signal drop in the upward bins; a bin below such drop is the lower most ground bin candidate. If ground bin candidates are more than five, the ground detection is not successful and therefore no ground bin is assigned for the respective observations [Lux et al., 2018].

    For illustration, we demonstrate the successfully detected ground bins and the cases without ground bin detected

using the official algorithm for the reference orbits from Table 2 shown earlier. As seen in Fig. 1, the cases with detected ground bins constitute a minor fraction of each orbit – of all observations, 12% (1 638 cases), 23% (3 119 cases), 28% (5 961 cases) and 18% (2 314 cases) contain detected ground bins for the reference orbits from 2018.09.10, 2018.11.30, 2019.01.11 and 2019.05.01, respectively. However, most cases with no ground bin detected originate from ocean areas with very weak water returns, manifesting a signal of very low magnitude (potentially – noise). According to our previous experience

[Labzovskii et al. 2023], more detected cases in winter are explained by the presence of sea ice over ocean. As we are interested in land surface signal, we applied the surface flag mask. To this end, we adopted the 'surface' parameter from the L1B data (see 'ground_wind_detection/measurement_ground_wind_detection/mie_measuremenet_ground_wind_bin') and recalculated the ground bin detection statistics for land LSR only. Among land observations, 33% (1 039), 36% (2 581), 64% (3 438) and 32% (1 321) profiles contain ground bin signal for the same reference orbits from 2018.09.10, 2018.11.30,

2019.01.11 and 2019.05.01, respectively. More detailed statistics on the number of observations containing ground bins, and are clear enough to be used for LSR retrieval, are presented in supplementary material (see Table S2 and Fig. S.4). According to the official algorithm of Aeolus ground bin detection, most cases, the highest ground bin is taken from #21, #22, #23 or #24 lidar bin (counted from the lidar instrument to the ground), depending on the local topography (see example of these statistics from Fig. S2 in the supplementary material). We remind that in the Aeolus processing chain, the #1 bin is

closest to the lidar detector, so the counting starts from the top in terms of atmospheric vertical profile [Flament et al., 2021].



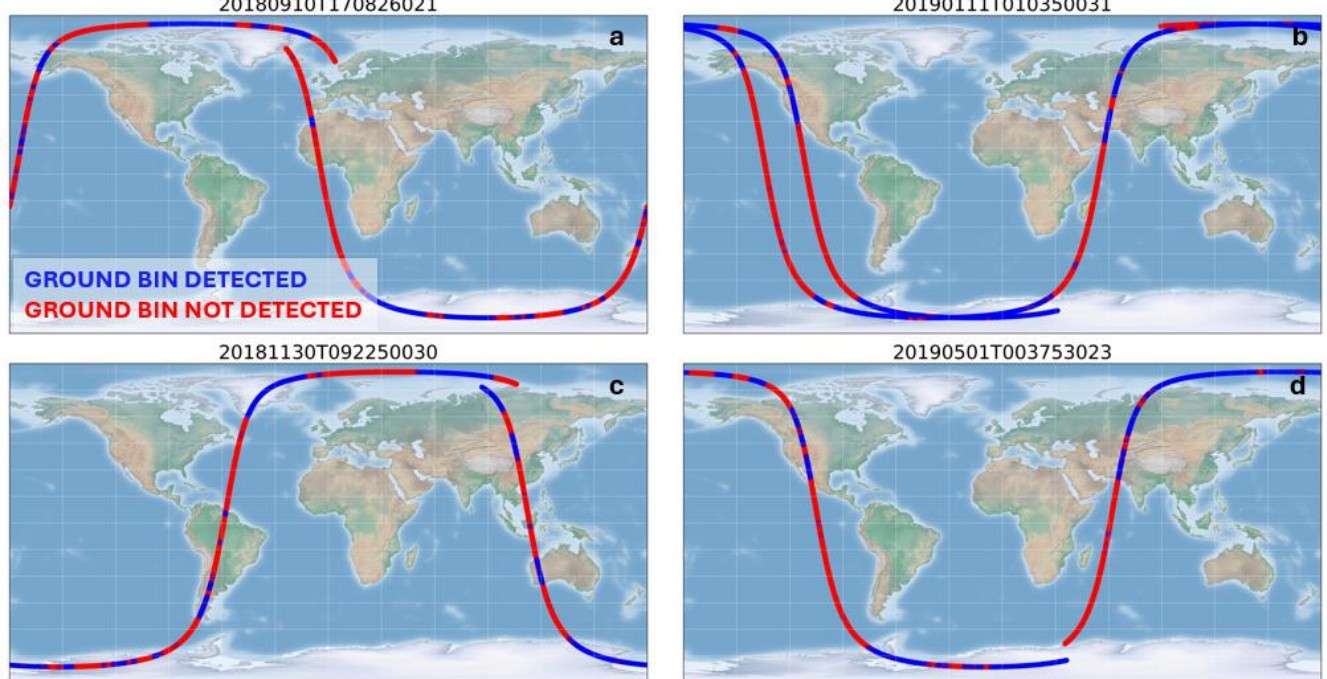

**Fig. 1.** Reference files used in this study to illustrate the methodology: 2018.09.10 (panel a), 2018.11.30 (b), 2019.01.11 (c) and 2019.05.01 (d).


In calculating LSR, we used all ground bin numbers marked as containing surface ('ground_bin_num') and integrated the attenuated backscatter across these surface bins. Here, the LSR is quantified through integrating the attenuated backscatter signal as previously indicated by CALIPSO-based studies [Josset et al., 2008; Lu et al., 2018]. To this end, we adopt the cross talk corrected attenuated backscatter signal (β) from the L2 AEL_PRO data using the Mie spectrometer data

only. The integrated signal represents the Surface Integrated Attenuated Backscatter (SIAB') that can be simply referred to as the uncorrected Lidar Surface Return (LSR'). Eq. 1 shows how we calculated the uncorrected LSR' through integrating attenuated backscattering (β) between the lowest bin, where ground is located (s_min) and highest bin where ground is located (s_max) according to the DEM information at the given range-bin-thickness (Δr) that takes into account the Aeolus pointing angle. In the supplementary material, more illustrative figures are provided, demonstrating the magnitude and

distribution of LSR' calculated by Eq. (1) in Fig. S1.

$$\gamma_{,} = \sum_{s\_min}^{s\_max} \beta\,(z_i)\Delta r(z_i)$$





(1)


### 2.2.2 Atmospheric correction of LSR

While utilizing lidar surface backscatter, it is crucial to develop a methodology that maximizes the extraction of useful information on surface reflectivity properties from lidar signals while taking into account the impact of atmospheric profile characteristics. Given the small field-of-view of ALADIN [Lux et al., 2018; Reitebuch et al. 2018], the LSR attenuation due to Rayleigh scattering can be simply corrected using the Beer's law (i.e. single-scattering). Within the atmospheric correction of Rayleigh signal, we first obtained the Rayleigh extinction coefficient ($\alpha_m$) profiles from the Aeolus L2A data.

In essence, these values are determined from the atmospheric density profile (derived from ECMWF forecast data). Eq. 2 below describes the calculation of total Rayleigh optical depth ($OD_{Ray}$). Then, we integrated the Rayleigh extinction coefficient between the surface and the Aeolus top altitude while accounting for the effects of a partially filled surface bin and optical depth above the Aeolus top-bin that is not considered in the total optical depth initially calculated (e.g., 'missing part' of the optical we referred to in Labzovskii et al. [2023] methodological description placed in the supplementary).


$$OD_{Ray} = \sum_{i_s}^{i_t} \alpha_m(z_i)\Delta r(z_i) - k1 + k2$$

(2)

where z is the altitude, $i\_t$ is the top range index, $i\_s$ is the surface range index and $\Delta r$ is the range-bin-thickness taking into account the Aeolus pointing angle. Note that $k_1$ and $k_2$ are two correction factors that need to be considered. The $k_1$ factor

accounts for the potential of over estimation in Eq. 2 due to the surface elevation situated above the lower boundary of the surface bin. This factor is calculated using the molecular extinction coefficient at the surface bin and the difference in the top boundary of the surface bin and the expected surface height according to the DEM information included in the Aeolus products used here. Secondly, the correction factor $k_2$ is required to alleviate the difference between the highest top bin of the Aeolus profile and the top of the atmosphere [Stephens, 1994]. We take into account the pressure at the top of the Aeolus

profile ($p_t$ [mb]) and the highest range gate altitude ($z_t$) [km], the Aeolus wavelength of 355 nm ($\lambda$) and the given ($\mu$), the cosine of the Aeolus off-nadir pointing angle (usually ~35°). As the Aeolus off-nadir pointing angle may differ depending on the location, the angle estimate is directly taken from the 'elevation angle' array of Aeolus data.



$$k_2 = \mu^{-1} \frac{p_t}{1013.25} \, 0.0008 \, \lambda^{(-4.15+0.2\,\lambda)} \, e^{\left(-0.1188 z_t - 0.0016 z_t^2\right)}$$


(3)

Next to the molecular attenuation, the attenuating effects from Aerosol and thin clouds must be taken into account for ensuring the cleanest LSR statistics, manifesting ground returns only, and removing all atmospheric effects. For this, we
utilized the AOD corresponding to the aerosol and thin cloud extinction profiles retrieved by AEL-PRO data [Donovan et al., 2023]. As ALADIN was a High-Spectral Resolution lidar (HSRL), , an extinction profile can be retrieved directly without assuming the lidar extinction-to-backscatter ratio profile [Shipley 1983]. Unlike elastic lidar based techniques, this theoretically allows for providing more accurate extinction coefficients. AEL-PRO uses both the pure Rayleigh and Mie attenuated backscatters as input for its retrievals. By applying a cost-function, the optimal-estimation approach determines
the likelihood of measurements given a specific forward model and our expectations. In brief, both AOD and RayOD estimates are both here used to calculate the corrected LSR signal ($\gamma$) at the original resolution of Aeolus (see Eq. 4 below).

$$\gamma = \gamma' \, e^{2(AOD+OD_{Ray})}$$


(4)

Although this is not a technical paper, exclusively dedicated to the LSR software description, we shortly illustrate the scheme of Aeolus LSR retrieval below for the convenience of the reader (see supplementary material, Fig. S3). The errors
behind LSR are calculated based on using instrumental uncertainties of input parameters of LSR equation (Eq. 4) in a simple error propagation formula. The idea is to understand how errors of LSR' (rooted sum of squares e.g. RSS of AB instrumental uncertainties for lidar bins with ground) and total aerosol optical depth (RSS of extinction errors along all lidar bins over ground) propagate into final LSR uncertainties. To this end, we assume that uncertainties in the variables are independent and that the partial derivatives are evaluated at the mean values of the variables so the contribution of both optical depth and
LSR' can be disentangled.

### 2.2.3 Additional processing of LSR data: quality flags and gridding
Our previous work has indicated that LSR can be excessively weak due to presence of strongly attenuative (or even
obscuring) features like heavy aerosol loading, thin or thick clouds [Josset et al., 2008; Hu et al., 2008; He et al., 2016]. Most



crucially, as shown in Labzovskii et al., [2023], the strength of the Aeolus LSR signal varies depending on the surface reflectivity characteristics. Thus, it is imperative to ensure that LSR comes from the surface and its magnitude is not altered by attenuation from unaccounted atmospheric features such as clouds or aerosols. Due to this, besides correcting the LSR' for aerosol and molecular atmospheric extinction, the effects of atmospheric features that can weaken or completely

attenuate the surface echo must be minimized (e.g., LSR signal-to-noise ratio is high). We repeated the quality control procedure based on the use of AEL-PRO L2 data [Donovan et al., 2023] used earlier in the first Aeolus LSR-focused study following Labzovskii et al. [2023]. Specifically, we calculated the percentage of attenuative features above the ground bin (which contains either attenuation or water/tropospheric cloud) with regards to the total number of Aeolus bins. The attenuative features are all those cases marked with codes 1 (Water cloud), 2 (Ice cloud tropospheric), 101 (Water cloud) and

<9999 (other attenuated feature flag) from AEL_PRO. A table with the codes of each atmospheric feature is included in the supplementary material (Table S.2). This quality control parameter has been previously denoted for LSR purposes as atmospheric quality flag or qflag for brevity. It ranges from 0% (no attenuative features over the ground bin) to 100%, whereas the latter means that all features over the ground bin are attenuated. We applied the most stringent filtering strategy by filtering out all LSR observations with qflag > 0. Finally, we filtered out all the LSR observations with AOD > 1.0

(calculated from the AEL_PRO integrated profile of extinction coefficients), thereby ensuring that observations that are attenuated by excessively hazy conditions are not included in the analysis. We refer to these resultant observations that passed the threshold mentioned above as to "clear" or "final" in this paper. The statistics on how many attenuative features have been filtered out are provided in the supplementary material (See Fig. S.4 and Table S.2). In short, although the AOD = 1.0 threshold might seem arbitrary, additional analysis on how various AOD thresholds affect the final selection of LSR

observations (and therefore 2.5 x 2.5 gridded LSR maps we describe below) yielded only very minor differences. This analysis is provided in the supplementary material (see Fig-s S4 and S5 alongside the corresponding paragraph).

      After filtering out high AOD cases and clouds, we gridded the final selection of LSR observations. The gridded estimates are needed to compare Aeolus LSR estimates to LER references at regional and global scales, while also understanding the prospects of the LSR product as a potential L3 climatology product. We averaged the LSR estimates for

each month by creating 2.5° x 2.5° geographical grids and populating these with Aeolus observations. We further applied spatial joined operation using the geopandas.sjoin function of the 'geopandas' package. In this way, we calculated monthly average estimates of LSR for each grid cell with the associated uncertainties (as 1 sigma of LSR during the month). We summarized all the steps for calculating scientific product from LSR used in this paper on Fig. 2 (from first to the last step, depicted by going from top block to bottom one). Note that the final LSR on a 2.5 x 2.5 degrees global grid are calculated

either for one month or for one year averages. In each case, we do mention which average we refer to in the beginning of the respective paragraph or in a form of remark.





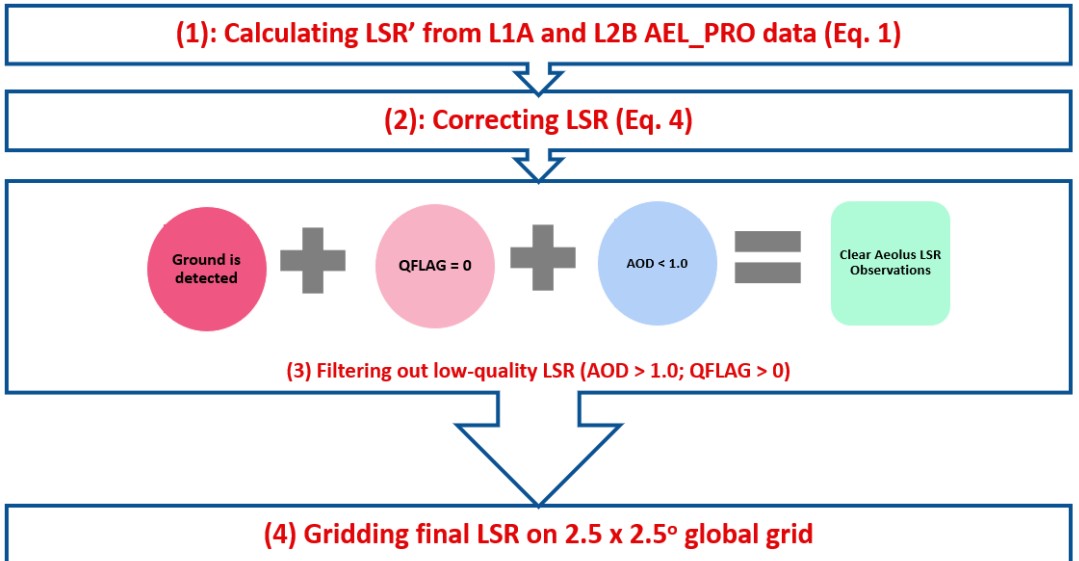

**Fig. 2** Methodology: From raw LSR to AOD-corrected LSR where steps are shown in chronological order to be done from top to bottom

Below, we show some examples with LSR vertical profiles from the reference orbits (Fig. 3). We plotted vertical profiles of attenuated backscattering (AB) from Aeolus observations with markers, signifying the ground detection (see red circles on Fig. 3). All these cases are taken from the reference orbit we described before with the index of observational point expressed over top of each subplot. The methodological framework shown in Fig. 2 is based on two types of observations from the LSR standpoint: attenuated observations (quality flag > 0) and clear sky observations (quality flag = 0). Attenuated cases are illustrated on the left side of Fig. 3 (in the red frame), where AB peaks above the ground either completely attenuated or weaker than the LSR we are interested in. Three attenuated cases over arctic waters (1), northern Canada (2) and arctic islands in Russia (3) from Fig. 3 exhibit some atmospheric peaks at ~1 500 m (qflag = 0%), 4 000 – 6 000 m (qlfag = 36%) and ~2 200 m (qflag = 15%) over the ground, respectively. Despite this, all ground detection bins occur at the highest signal peaks at the lower altitudes, where ground was detected by the official detection algorithm. To ensure the clearest LSR statistics all such potentially attenuated cases are filtered out from the final analysis since these attenuative features would still weaken the surface echo. As mentioned, quality flags here indicate how much attenuative features (in %) were detected over the ground bin from the total number of lidar bins. It is worth noting that despite having highest quality flag (0%), case 1 from Fig. 1 exhibits some attenuation peak around 2000 m. This example illustrates the importance of filtering out high AOD cases, performed at the step 3 of our methodology, shown in Fig. 2, which can otherwise remain unaccounted for. Among the unattenuated or clear cases included in our final analysis, one can notice profiles over the Ural Region (4), Western U. S. (5), Antarctica (6) and Indian Ocean (7) in Fig. 3. As seen, all land cases exhibit very strong and strong ground returns at different altitudes with 2 bins detected over Western U. S. case (due to topography) and 1 ground



bin detected in other cases. The altitude of the ground bin varied from ~0 m in Ural and Indian Ocean to >2 000 m in high altitude cases of Western U. S. and Antarctica. Unlike in the Labzo-23 method, clear LSR observations over oceans are scanty. The official ground detection algorithm is missing ground bin over oceans due to excessively weak signal, considered as noise.


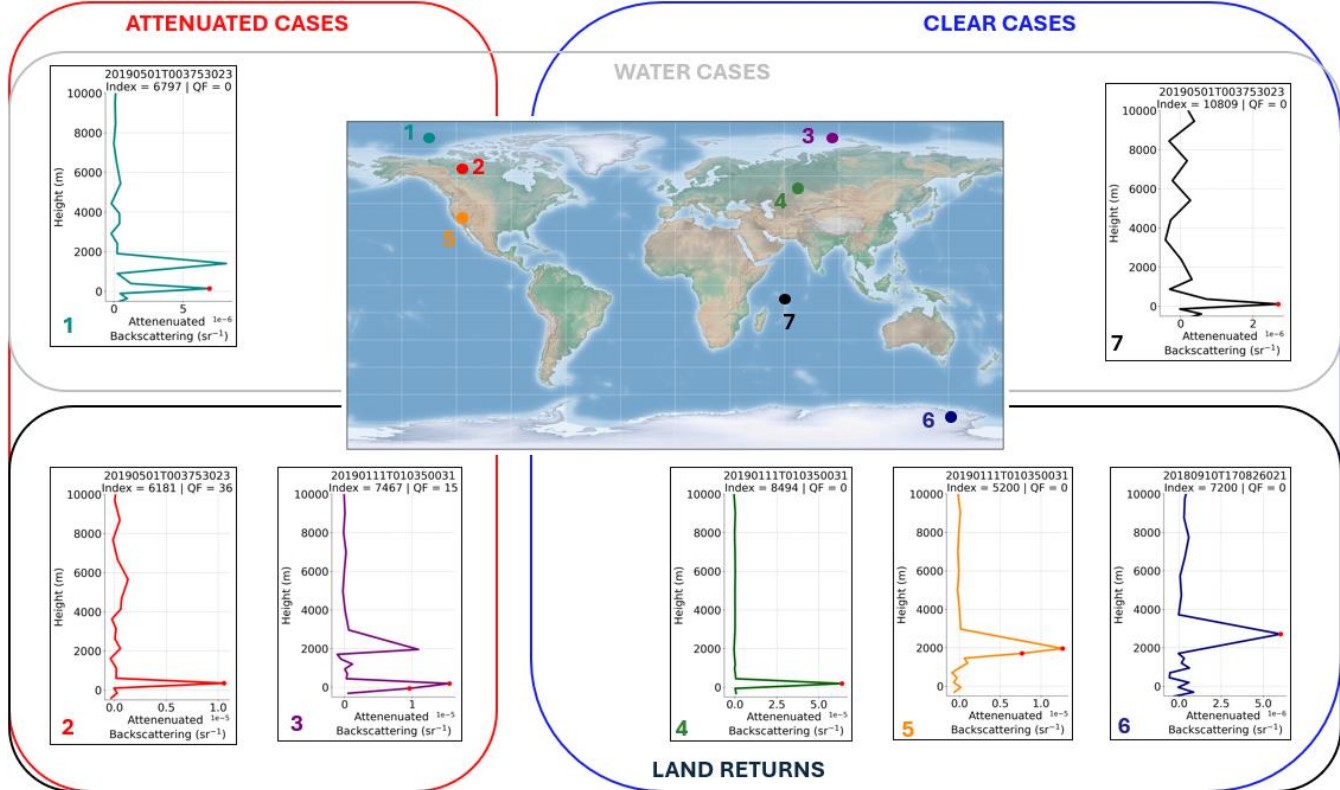

**Fig. 3** Examples of vertical profiles of attenuated backscattering (ATB) for four reference orbits selected for the paper. Red frame: attenuated cases (qflag > 0), blue frame: clear cases (qflag = 0%), grey frame: cases over ocean (surface flag indicates water), black frame: cases over land (surface flag indicates land). 1 – Arctic waters, (2019.05.01), 2 – Northern Canada (2019.05.01), 3 – Arctic Island in Russia (2019.01.11), 4 – Ural Region (2019.01.11), 5 – Western U. S. (2019.01.11), 6 – Antarctica (2018.09.10), 7 – Indian Ocean (2019.05.01). Note that index of observations illustrates the number of observations in this particular orbit. Reference of the orbit according to L1B format is provided as well.




Fig. 4 illustrates the reference analysis and histograms of LSR distribution depending on the surface type, namely, depending on the surface flag. The distribution of LSR for every reference orbit selected earlier for three types of surfaces: land, water and water in low latitude regions (black, blue and magenta colours in Fig. 4, respectively). We assume that the waters

between -35 and 35 degrees should be ice-free since no ice flag was included in the Aeolus L1B data. For the four selected orbits, the LSR ranges from 0.0 to ~0.6 sr$^{-1}$. Plausibly, the maximum LSR values are limited by lower levels in the September orbit (low amount of snow in northern hemisphere, low amount of ice in the southern ocean) with a maximum of ~0.4 sr$^{-1}$. All land LSR distributions are bimodal, with a weaker LSR peak at < 0.1 sr$^{-1}$ and a stronger LSR peak at 0.2 sr$^{-1}$. These differences are explained in the results section below. The water LSR peaks are either bimodal, e.g. September 2018,

or unimodal, whereas the latter pattern is explained by the low returns from sea ice surfaces. As mentioned in the text placed under Eq. 4, LSR errors were calculated using error propagation considerations taking into account optical depth and uncorrected LSR estimates. The errors for the example orbits, mean error estimates are 19±9% for 2018.09.10, 16±5% for 2018.11.30, 14±5% for 2019.01.11, 21±7% for 2019.05.01 orbits. There are large differences between land and water LSR mean error estimates (16-23% and for land and water, respectively). The larger errors over land is explained by higher

relative contribution of AOD error in the error propagation procedure we applied (59–69% depending on the orbit). In other orbits, errors are very similar, not shown here, but will be available upon the publication of the official LSR dataset during Aeolus Phase-F stage. Notably, once all higher latitude regions are clipped from the analysis (magenta bars), only a very low number of strong returns (< 100 cases) remain, compared to land LSR statistics. This difference is explained by the difference in the way how ground bin is detected using the official method, compared to Labzo-23 method we mentioned

earlier.





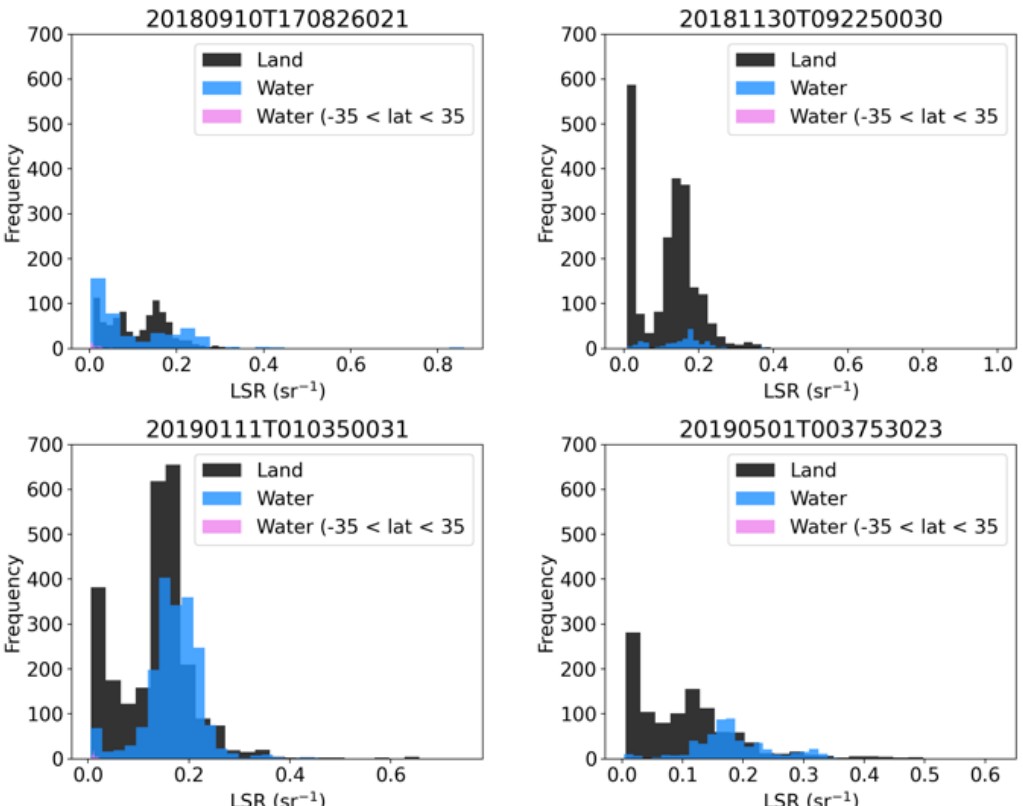

**Fig. 4** LSR histogram distributions for four reference orbits on 2018.09.10 (a), 2018.11.30 (b), 2019.01.11 (c) and 2019.05.01 (d) showing land (black), water (blue) and water outside high latitudes regions assumingly without ice (magenta)


## 3. Results

Our analysis cover several aspects of the LSR retrieval, such as a demonstration of the LSR distributions across four reference orbits, examination of LSR global distribution at 2.5 x 2.5 gridded average level, LSR evaluation versus LER references (GOME-2 and TROPOMI) for four references, for aggregated Aeolus orbits and for gridded levels. On top of that, we evaluate the sensitivity of Aeolus LSR to land cover characteristics such as snow and vegetation cover proxy, thus examining two hypotheses suggested earlier [Labzovskii et al. 2023] on the strong sensitivity of LSR to snow cover and moderate sensitivity of LSR to vegetation cover.


### 3.1 Examination of LSR regional patterns: orbits, monthly gridded estimates and seasonal gridded estimates



For illustration purposes, we demonstrate LSR distributions across the four mentioned orbits below in Fig. 5. Note that only the clearest cases are shown (qflag = 0%). Several patterns are visible. First, there are less observations that are deemed
"clear" in this official ground bin detection method from oceans, compared to the Labzo-23 approach to detect surface bins. In many cases, the surface bin here is simply missing, while in the Labzo-23 method such bins were considered to have very weak signal. However, there are some clear ocean surface returns such as those over Eastern Pacific on 2018.09.10 (Fig. 5a) or several clear ground bins over Indian Ocean on 2019.05.01 (Fig. 5d). As expected, the strongest and most continuous sets of LSR observations are retrieved over high latitudes, covered by white surfaces of snow or ice [Taskanen and Manninen
2007; Weiler, 2017]. The prevalence of LSR returns from high-latitude regions using the official ground bin detection algorithm is plausible and had been already mentioned by Weiler et al. [2021]. Yellow-coloured observations from Fig. 5, manifesting strongest LSR returns from snow/ice surfaces (> 0.16 sr$^{-1}$) are visible over Arctic and Antarctic regions in all four analysed orbits. Over land outside high latitudes, there are abundant LSR observations with highly variable LSR magnitude. In most cases, land LSR for these four referenced orbits varies from ~0.05 to 0.16 sr$^{-1}$, but more detailed statistics
are described in this section.

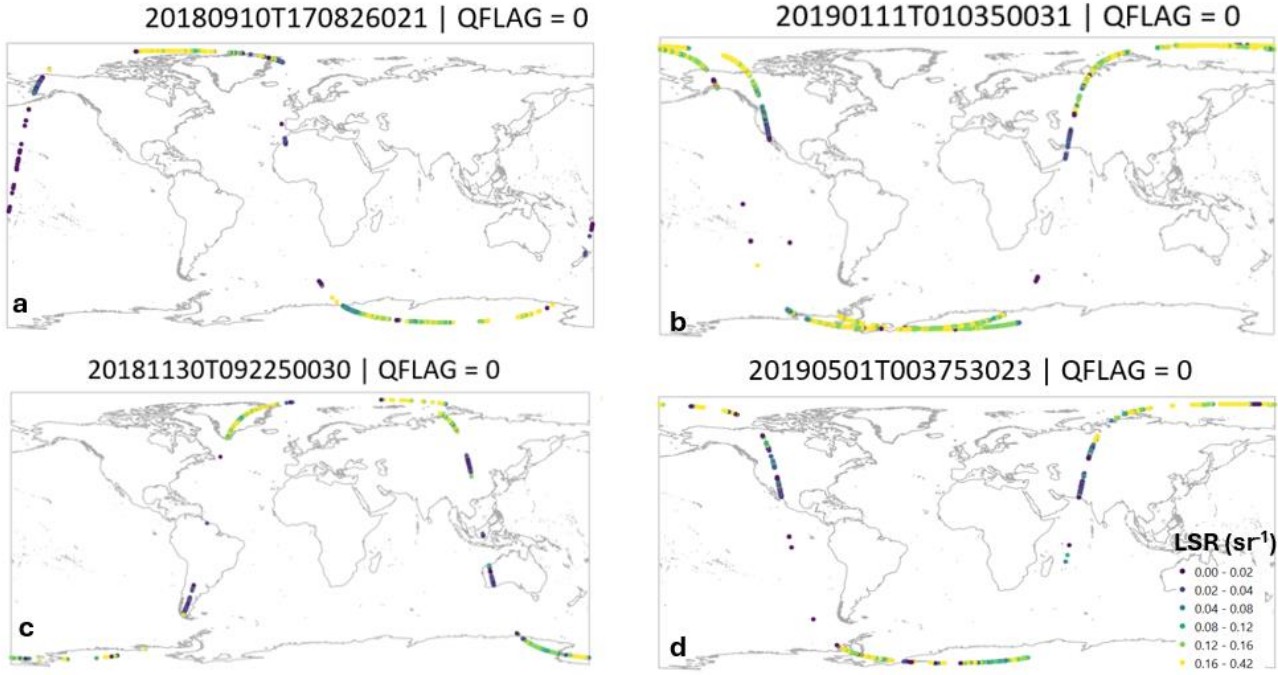

**Fig. 5** LSR distributions across four reference orbits on 2018.09.10 (a), 2018.11.30 (b), 2019.01.11 (c) and 2019.05.01 (d). Only the highest quality flag (qflag = 0) observations are included in this analysis.






Since Aeolus LSR has been previously shown to reasonably reflect several land cover-related gradients on the map such as: water – land, vegetation – arid, no snow – snow gradients, we used a clustering method to classify the LSR signal for better illustration purposes. To this end, we used natural breaks-based clustering of LSR data for plotting (e.g. Jenks clustering method) for identifying breakpoints between different clusters of LSR data [Sadeghfam et al., 2016]. The method minimizes

the average deviation (e.g. variance as well) of each class from its respective mean, concurrently maximizing the divergence of each class from the means characterizing the other classes [Jenks, 1967]. Note that we simply applied this approach for clustering and visualization purposes without intention to disentangle physical differences behind reflectivity patterns of different regions.

Fig-s 6-8 below show the LSR distribution on two different resolutions including the original Aeolus resolution

(LSR per sounding), shown at the left panel of each figure and the gridded mean LSR estimates at 2.5$^{\circ}$ x 2.5$^{\circ}$ grid cell resolution. We plotted not only gridded LSR estimates, but also the observation-resolution LSR estimates to give a visual insight into the data abundance behind the gridded estimates. Fig. 6 demonstrates that most signals with the detected surface according to the official algorithm were found over land in autumn. In line with Fig. 5 four-orbit statistics, the ocean bins were detected only for some regions like the Pacific Ocean (September, October and November 2018) and some other

scattered, less spatially distinct regions like the Mid-West Atlantic. As expected, in most cases, there is a distinct gradient between strength of LSR over water returns (shown in dark blue reflecting LSR < 0.018 sr$^{-1}$) and land returns (in most cases > 0.018 sr$^{-1}$). Moreover, agreeing with our first Aeolus LSR-focused work [Labzovskii et al., 2023], the strength of the LSR signal is visibly enhanced over the areas, covered by snow and ice. Snow/ice-covered areas are seen by orange-red colours in Fig. 6 (LSR > 0.121 sr$^{-1}$) and this cluster plausibly moves southwards from September to end of November in the Northern

Hemisphere (see northern Canada and Russia on Fig. 6 (panels d, f)). In line with the Labzo-23 method, some LSR gradients discerned from the currently applied official ground bin detection algorithm are visible here such as water – land, snow – no snow gradients. Interestingly, the differences between arid and vegetated areas reported in our previous work, are not salient here.





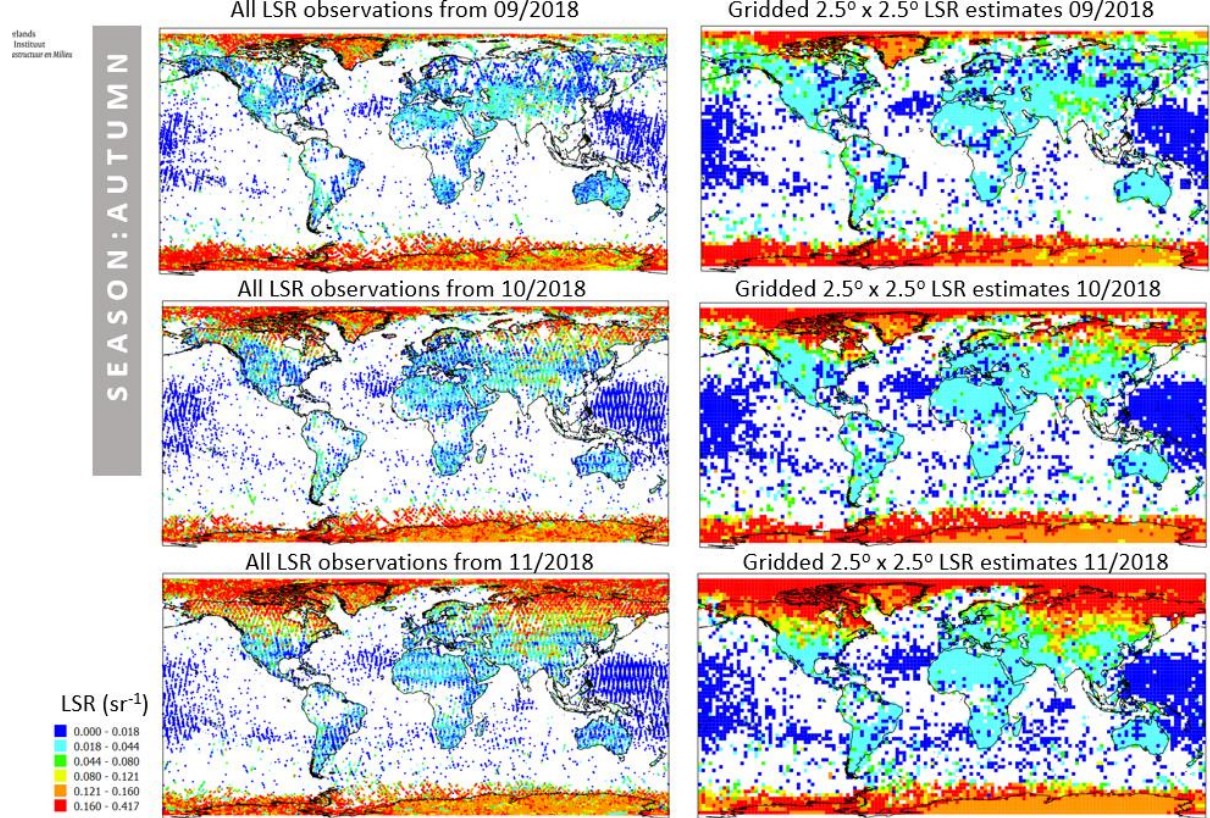


**Fig 6**. Panels a, b, c: all clear (qflag = 0, AOD < 1.0) LSR observations for September, October and November 2018, respectively. Panels d, e, f; gridded LSR estimates at 2.5º x 2.5º resolution based on the observations from the left for same months

Fig. 7 illustrates the LSR distributions in the winter of the FM-A period. It is obvious that LSR is very sensitive to snow cover changes in the northern hemispheric winter, whereas very strong LSR returns are being registered over major parts of Canada, Northern Russia, Central Asia and the Himalaya. Moreover, large numbers of strong signals were discerned over water near Antarctica and the Artic regions, indicating the presence of sea ice. Interestingly, the number of strong water returns is at their minimum in February 2019. Moreover, some regions like Mid-West Africa or Amazon are missing from

the final gridded estimates. Some weakening of LSR signal over these regions, especially, over Mid-West Africa, is attributed to the diminishing of surface signal due to high AOD [Labzovskii et al., 2023]. Alternatively, this phenomenon could be caused by extremely weak LSRs, which, according to the official surface bin detection approach, are assumed to lack any surface signal. We underline that Mid-West Africa and the Amazon are being most heavily influenced by biomass burning [Randerson et al., 2012] and tropical cloud convection processes [Chakraborty et al., 2019] among other geographic

areas. Fig. S4 of the supplementary material indicates that such atmospheric conditions lead to a dearth of clear LSR



observations over the region even if one lifts the clear LSR threshold to "AOD = 1.5" in the quality assurance procedure, shown in Fig. 2.

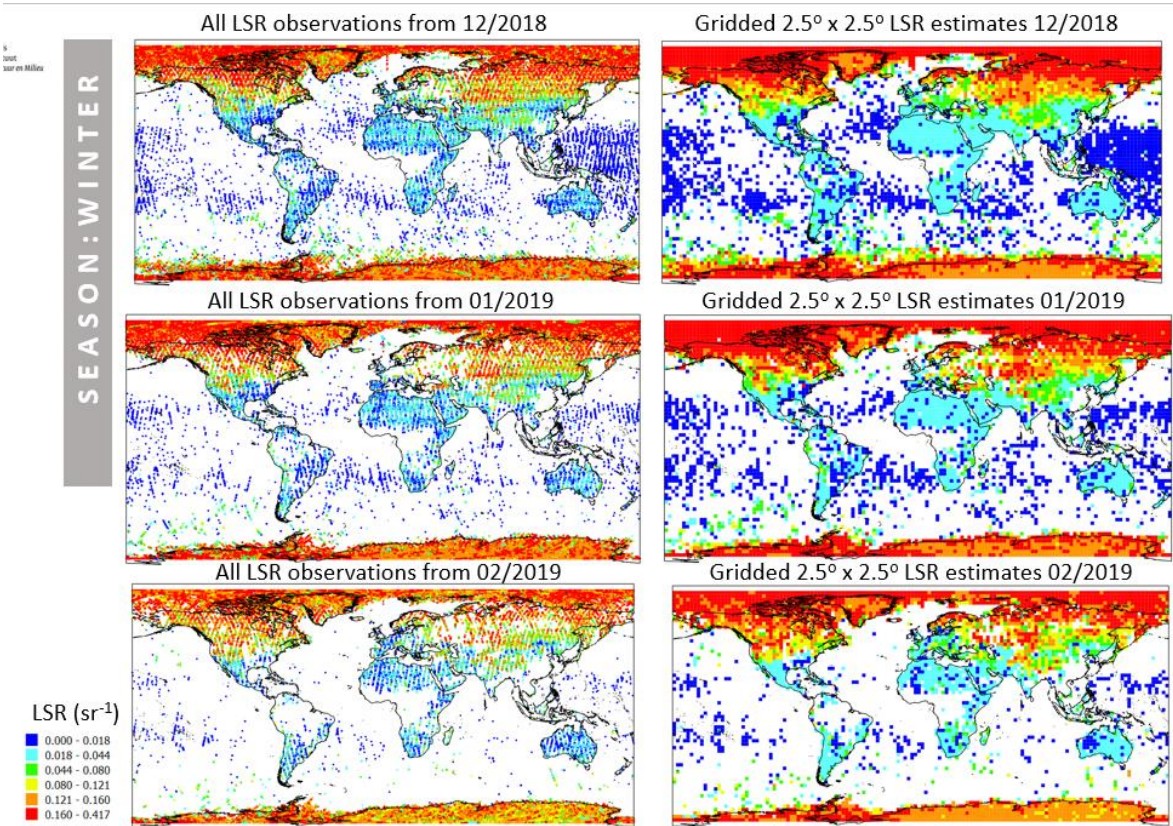

**Fig 7**. Panels a, b, c: all clear (qflag = 0, AOD < 1.0) LSR observations for December 2018, January 2019 and February 2019, respectively. Panels d, e, f; gridded LSR estimates at 2.5° x 2.5° resolution based on the observations from the left for same months

Fig. 8 below shows the northern hemisphere spring LSR distributions. A line of snow retreat towards the north is distinctly visible in the northern hemisphere as seen by the shift of the orange-to-red LSR cluster northwards. While snow-related clusters nearly disappeared from the northern hemisphere in May 2019, strong LSR signal remained over the Arctic, perhaps indicating a localized peak in sea ice seasonality in the region. Moreover, there is a noticeable weakening of the signal over the entire land regions in the northern hemisphere, manifested on the map as the emergence of dark blue-coloured clusters similar to water in magnitude. This is potentially related to the greening of vegetation during northern hemisphere growing season as indicated by passive remote sensing studies [Tilstra et al., 2017] and our previous lidar-based LSR work [Labzovskii et al., 2023]. From a land reflectivity perspective, the weakest UV returns are registered over densely vegetated,



i.e., green areas. More unexpectedly, the distribution of the detected ocean surface returns changed in spring 2019. Specifically, the LSR formed two longitudinal bands near the tropics, which reach their respective peaks in area in March – April 2019. These ocean areas are the so-called south hemisphere gyres, where the concentration of chlorophyll is very low, near surface wind speeds are low and ocean mixing is weak [Morel et al., 20211]. It should be noted that we do not have sufficient empirical arguments to support this hypothesis though and such analysis is outside of the scope of this paper.

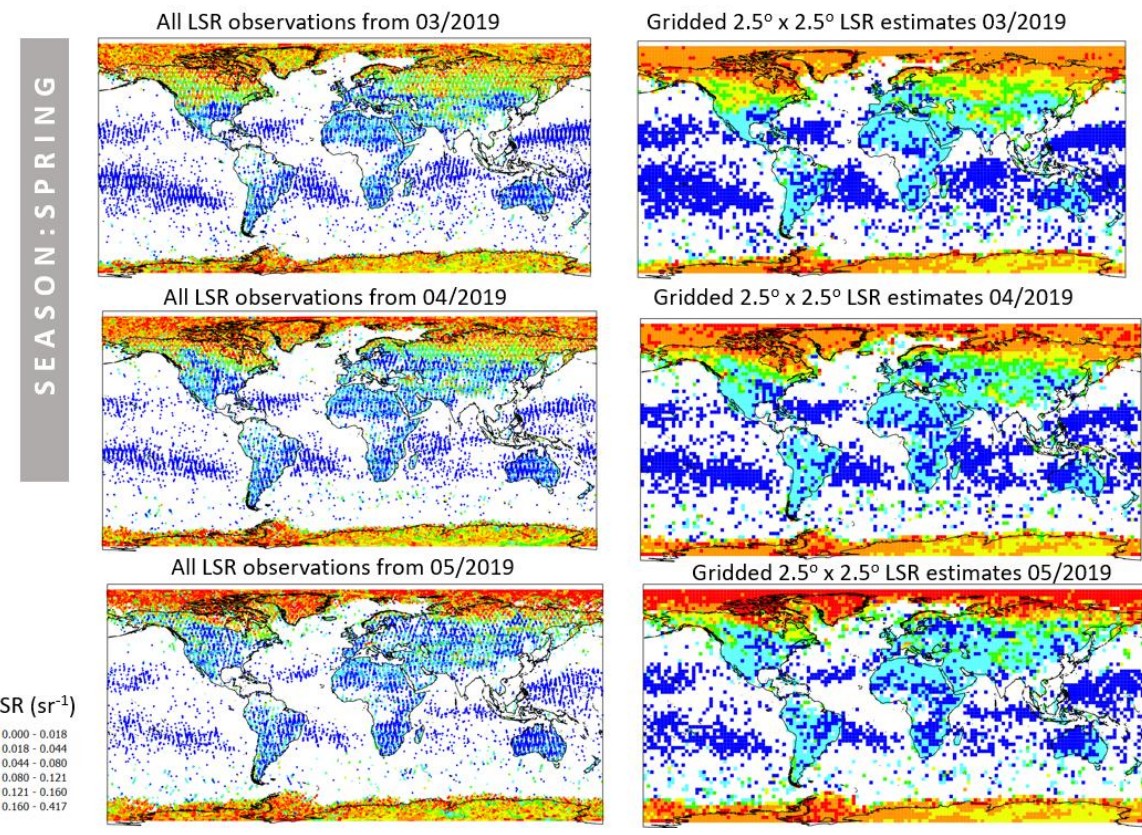

**Fig 8**. Panels a, b, c: all clear (qflag = 0, AOD < 1.0) LSR observations for March, April and May 2019, respectively. Panels d, e, f; gridded LSR estimates at 2.5° x 2.5° resolution based on the observations from the left for same months

To finalize the global seasonal analysis of LSR, we also visualized the LSR mean gridded estimates for the entire FM-A period and for each season separately in Fig. 9. Interestingly, LSR exhibited several distinct gradients like (1) land – water, (2) snow – no snow, (3) snow – ice. To remind, this coloring reflects the Jenks optimization clustering result, described earlier. In particular, the gradient between land and oceans is best visible in the difference between light blue (0.018 – 0.044 sr[-1]) and dark blue (< 0.018 sr[-1]) colours, respectively. Over land, LSR can strongly vary with the weakest signals, similar to water returns, observed mostly over Northern Hemisphere during high productivity seasons. In general, LSR over land varies from 0.018 to 0.121 sr[-1] outside very high latitudes or highland areas, where snow can be formed (see the yellow clusters in Fig. 9 over North Siberia, North Canada, Tibet and Pamir mountains for example). On one hand, the difference between arid





and vegetation areas is unexpectedly subtle on such maps. On the other hand, orange clusters indicate the areas where snow

can be found (0.12 – 0.16 sr⁻¹), while red-coloured areas are located over high-latitude seas/oceans, representing sea ice formation areas with the strongest LSR of > 0.16 sr¹. This is an interesting finding as the LSR magnitude over sea ice is stronger using the current official ground bin detection method, compared to the Labzo-23 method, which has previously yielded the highest gridded values of ~0.10 sr⁻¹ over Greenland and Antarctica. Moreover, the difference between ice-covered ocean areas and snow-covered areas was also not distinct in the Labzo-23 approach. This is an interesting result that

can be explained by the ability of the official ground bin detection approach to register only strong returns. In other words, while Labzo-23 gridded statistics would be based on omnipresent weak returns from water alongside occasional cases of sea ice from high-latitudes, the current approach yields only strongest returns as the weaker ocean returns are not considered as ground bin detections here. See noticeable scarcity of any weak ocean signals over Southern Ocean – they are simply missing on global statistics in Fig-s 6–8.


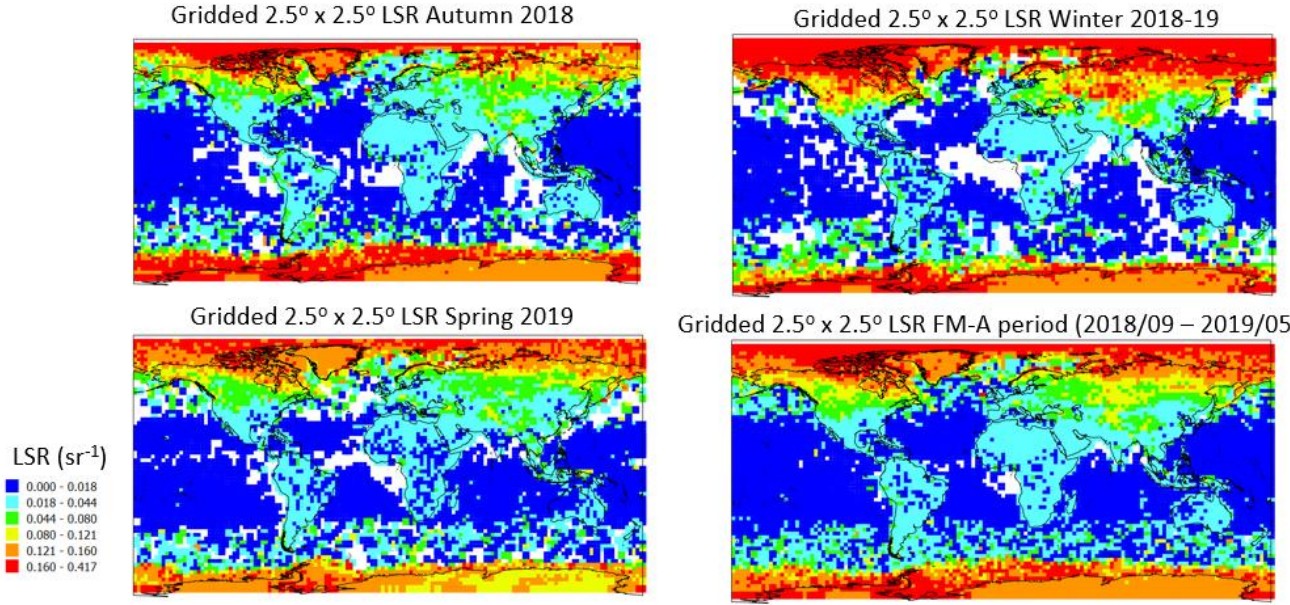

**Fig 9**. Seasonal LSR means at 2.5º x 2.5º  grid for Autumn 2018 (a), Winter 2018-19 (b), Spring 2019, (d) entire FM-A period from 2018-09 to 2019-05. Note that we refer to Northern Hemisphere seasons here.


**3.2 Evaluation of LSR retrievals versus LER references (TROPOMI and GOME-2)**





Our previous work had demonstrated unexpectedly high agreement between LSR and reference UV reflectivity datasets, namely LER climatologies from TROPOMI and GOME-2 [Labzovskii et al., 2023; Tilstra et al., 2017]. As our analysis had
been limited to the global comparison of gridded 2.5° x 2.5° mean estimates and regional averages across >30 arbitrarily selected regions, here we extended the analysis further.

### 3.2.1 LSR vs LER references for four selected orbits

First, we evaluated the agreement between LSR and reference LER estimates for the four reference orbits we selected for the
analysis earlier including: 2018.09.10, 2018.11.30, 2019.01.11 and 2019.05.01, as shown on Fig. 10 and Table 2. We sampled LER values from multi-year climatologies to each Aeolus observation for each orbit of interest and unveiled the following agreement patterns. First, there are two distinct populations of LER including very weak reflectivity (< 0.2) and very strong reflectivity (> 0.8) dominating the statistics for every orbit (Fig. 10). Second, both LSR-LER$_G$ and LSR-LER$_T$ comparisons exhibit high agreements with varying correlations for every orbit. This agreement varies depending on the orbit
and whether we compared LSR with TROPOMI or with GOME-2 estimates. In short, over all surfaces, correlation coefficient (r) ranges from 0.55 in 2019.05.01 (LSR-LER$_G$ comparison) to 0.77 in 2018.11.30 (LSR-LER$_T$ comparison). The agreement between LSR and LER, except 2018.09.10 orbit, is driven by the agreement over land, as indicated by Table 2.

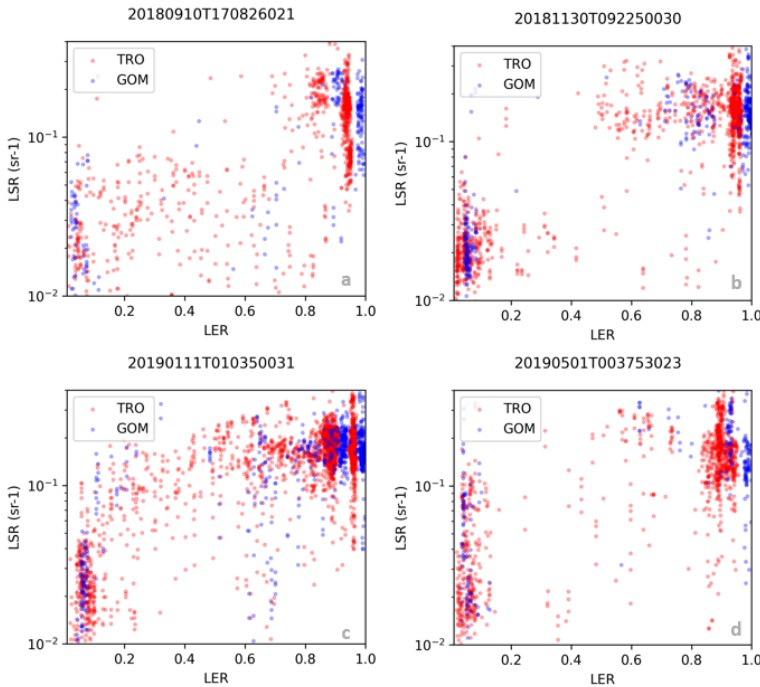

**Fig. 10** Aeolus LSR vs LERG (blue) and vs LERT (red) for four reference orbits of 2018-09-10 (a), 2018-11-30 (b), 2019-01-11 (c), 2019-05-01 (d). LSR axis-y is log-scaled.



**Table 2** Aeolus LSR vs LERG (blue) and vs LERT (red) for four reference orbits of 2018-09-10, 2018-11-30, 2019-01-11, 2019-05-01. Second and third columns: all observations; columns with marker 'l' – land; columns with marker 'w' – ocean/waters

| date | TRO-AEL | GOM-AEL | TRO-AEL_l | GOM-AEL_l | TRO-AEL_w | GOM-AEL_w |
|------|---------|---------|-----------|-----------|-----------|-----------|
| 2018.09.10 | 0.71 | 0.57 | 0.77 | 0.59 | 0.70 | 0.65 |
| 2018.11.30 | 0.74 | 0.77 | 0.76 | 0.79 | 0.08 | 0.08 |
| 2019.01.11 | 0.61 | 0.65 | 0.60 | 0.64 | 0.58 | 0.58 |
| 2019.05.01 | 0.55 | 0.64 | 0.52 | 0.60 | -0.13 | 0.41 |

Table W.1 Correlation coefficients in the comparison between Aeolus LSR (AEL) and reference LER estimates from GOME-2 (GOM) and TROPOMI (TRO) for all surfaces, land only ('l') and water only ('w').

### 3.2.2 LSR vs LER references for monthly aggregated orbits during the entire FM-A period

We further analysed the LSR-LER agreement for monthly aggregated orbits of Aeolus during the FM-A period. By monthly aggregated orbits we refer to all clear LSR observations per month merged into one dataset for statistical analysis. For each clear LSR observation, we sampled corresponding LER climatology points from both TROPOMI and GOME-2. Fig-s 11 and 12 show the comparison of Aeolus LSR versus two LERs references: LSR-LER$_G$ and LSR-LER$_T$, respectively. At this finest-scale observational level, we identified a weak-to-moderate agreement between LSR and LER estimates. In case of GOME-2, the highest agreement was found in September 2018 (r = 0.64) and the lowest in February 2019 (r = 0.44) showing no distinct seasonal patterns. Moreover, the LSR-LER$_T$ comparison yielded higher agreement with r > 0.60 for any month (with the highest agreement in September: r = 0.77 and the lowest agreement in November: r = 0.61). The lack of linear agreement at regional level can be related to different factors. First, there is a sigmoid-alike behavior of LSR across different land types with nearly exponential growth of LSR towards the most strongly reflecting regions – snow-covered areas [Labzovskii et al., 2023]. This effect can be seen by strong LSR "hot spots" on Figs. 11 and 12, which are elongated along y-axis, thus is indicating a higher sensitivity of Aeolus to snow. Second, this comparison is limited by strongly bi-modal distribution of LER with most values either distributed at low UV reflectivity range (< 0.20) or high reflectivity range (> 0.80), as we noted earlier from Fig. 10.

Indeed, the relationship between LSR and LER is far from being linear and therefore Pearson correlation agreement metrics would be inevitably skewed towards lower agreement metrics. One can assume linear association between LSR and LER by seeking for a correction factor. We performed such experimental attempt by applying different power law functions to LSR. Specifically, we applied different power law coefficients (l) in a simple power law equation ($\gamma^l$) by seeking the highest linear correlation between LSR and LER. We found that by correcting LSR through applying $\gamma^{0.1}$, we achieved the



strongest positive correlation between LSR and LER. In this case, the correlation between LSR and GOME-2 is increased r = 0.63 – 0.77 and for LSR vs TROPOMI is increased to r = 0.65 – 0.81 (depending on month). Overall, given no prior indications that LSR-LER agreement should be precisely linear, such agreement can be deemed generally high and promising, but no conclusions on the  physical relationship between these parameters can be made based on these statistics.






**Fig. 11** Global LSR vs LER (GOME-2) density plots for each month of the study period on the observational scale






**Fig. 12** Global LSR vs LER (TROPOMI) density plots for each month of the study period at the observation scale.

**3.2.1 Global agreement between LER and LSR**





We further evaluated the LSR ability to represent the surface reflectivity characteristics in the UV at global scales. To this end, we compared 2.5 x 2.5 degree gridded mean estimates of LSR versus both $LER_G$ and $LER_T$ (Fig. 13). In particular, each point in Fig. 13 represents the averaged LSR (or LER) estimate, corresponding to one grid cell shown in Fig-s 7-9 (FM-A period). We found a very good agreement between LSR and LER estimates for Aeolus-TROPOMI and Aeolus-GOME-2

comparisons, with r = 0.92 and r = 0.90, respectively. The very high agreement shows some improvement on the LSR-LER agreement, compared to our previous work [Labzovskii et al., 2023] which had reported (r) correlation coefficients with TROPOMI (r = 0.89) and GOME-2 (0.62) for the FM-A period. Since we used exactly the same dataset for LSR and GOME-2 (TROPOMI LER was updated to version 2.0), this improvement can be explained by the change of the methodology in (a) surface bin was detected or not for a certain observation and (b) assumption on how many bins contain

the surface information. Most likely, as indicated by Fig-s 7-9, the current official Aeolus ground bin detection methodology implies that not every observed profile contains surface backscatter returns. This results in many ocean surface returns, with the weakest LSR signals being filtered out from the analysis. This effect seemingly further improves the overall agreement between the datasets. This explanation is plausible because it is already known that LSR agrees well over LER mostly over land, not over water [Labzovskii et al., 2023]. We discuss these differences in short further in the paper.


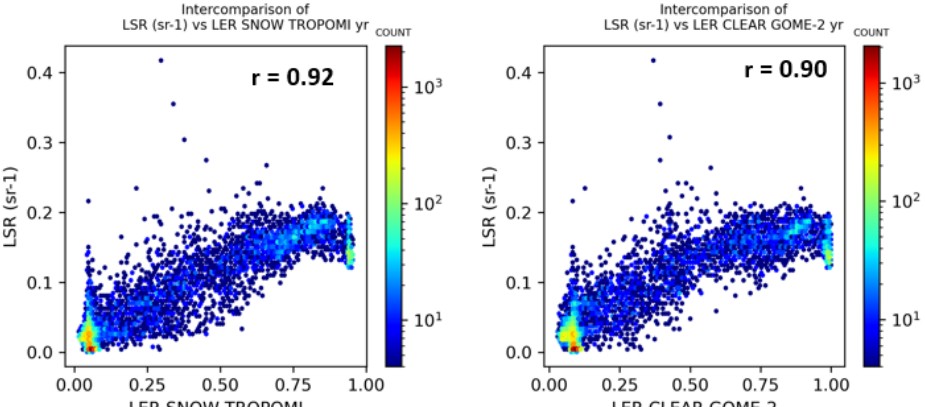

**Fig. 13** Results: Monthly gridded averages of LSR and LER for 2.5º x 2.5º grid cell resolution for FM-A period. Panel a: LSR vs TROPOMI LER; panel b: LSR vs GOME-2 LER.

**3.3 Region-specific analysis of LSR and LER**

Furthermore, we analyzed the regional agreement between LSR and LER. To remind, a promising agreement between LER regional monthly averages and corresponding Aeolus averages in arid regions like Sahara with low reflectivity variability during the year has been previously reported [Labzovskii et al. 2023]. In other words, very weak annual reflectivity variability of poorly-vegetated areas, that had been previously thought to be insignificant, has been detected by Aeolus LSR

in the September 2018 – September 2019 period. Since the prior seasonal analysis had been limited to the Sahara only, we




extend the seasonal analysis by incorporating > 30 regions within. These regions had been previously selected in Labzovskii et al. [2023] to represent typical geographical conditions from an ecosystem perspective. This region-specific analysis illustrates seasonal LSR-LER comparisons at monthly scales for each representative region from ecosystem point of view (Fig. 14) and correlation table between LSR and LER for each selected region is shown (Fig. 15). The regions in Fig. 15 are

ranked based on the TROPOMI-GOME-2 agreement in descending order with (red color – higher agreement).

In geographical terms, the highest agreement between LSR and LER was found in the regions where snow occurrence is common in winter (Fig. 15), thus forming white surfaces that strongly reflect UV light [Tanskanen and Manninen, 2007]. For example, high correlation coefficients of 0.87, 0.88, 0.86 and 0.75 were discerned between LSR and $LER_T$ in Eastern-Central Eurasia, Scandinavia, North Siberia and Northern Canda, respectively (in the case of LSR-$LER_G$

they were 0.89, 0.85, 0.79 and 0.72, respectively). Fig. 14c illustrates an example of Scandinavia, where both LSR and LER are sensitive to emergence of snow with reflectivity peaks in December – January. In terms of magnitude, LSR goes up to ~0.15 $sr^{-1}$ in January 2019, corresponding to ~0.3 in terms of LER. This one-peak curve is evident for both the LER and LSR estimates except in Northern Canada, where LSR decreases at a faster rate in comparison to LER estimates at the end of winter. Interestingly, for year-round ice-covered regions the agreement is less obvious with high agreement for Central

Greenland on one hand (r = 0.63 and r = 0.88 for LSR-$LER_G$ and LSR-$LER_T$ comparisons, respectively), but lower agreement over Antarctica on the other hand (r < 0.10 for both Aeolus-GOME-2 and Aeolus-TROPOMI comparisons). Note that LERG and LERT both yielded moderate agreement (r = 0.67 and 0.51) to each other over Antarctica and over Central Greenland, respectively. The reason behind the lower agreement is because LER estimates show very low variability throughout the year over these regions, while Aeolus detects several LSR changes in March – April 2019 in Central

Greenland (see supplementary material; Fig S. 7). It is hard to compare these dynamics with existing reference seasonal data on ice in Antarctica even qualitatively because most studies address ice extent [Parkinson, 2014] in the region, not snow cover, to which LSR is most sensitive to. Another reason can be the fact that Aeolus stands out with unprecedented coverage of these high latitudes with many observations in polar night and no issues arising from solar zenith angle, typical for passive remote sensing instruments [Tilstra et al., 2017]. Moreover, LER is a multiyear average and is filled in with constant with

detection is not successful, while Aeolus provides direct measurements of unidirectional reflectivity.

In arid and semi-arid regions, the agreement between LSR and LER is generally high for most regions with a few exceptions. Note that we labelled the regions semi-arid like Mongolia [Han et al., 2014] not based on the conventional ecosystem classification, but on the possibility of snow occurrence in these generally arid regions. Like in Labzovskii et al., [2023], a very good agreement between LSR and LER over Sahara is discerned despite the very weak reflectivity variability

identified (Fig. 14a). We have previously discovered a rather surprising sensitive of Aeolus to reflectivity changes in the Sahara desert [Labzovskii et al., 2023]. Importantly, here we confirm these findings by applying the official method of bin detection and extending our analysis to other arid regions, where vegetation changes are minimized. Specifically, for LSR-$LER_T$, good-to-very good agreement was found over all arid regions including Middle East (r = 0.81), Sahara (0.86), Iran (0.70) and also all semi-arid regions including Mongolia (0.90), Central Asia (0.96) and Arid U.S. (0.97). For LSR-$LER_G$





comparison, we found good agreement only for Middle East (0.80) among arid regions, but for all semi-arid regions
including Mongolia (0.91), Central Asia (0.96) and Arid U.S. (0.90). For semi-arid regions like Mongolia, shown in Fig.
14b, a one-peaked curve with the maximum in winter indicates that the agreement is driven by the presence of snow, which
manifests the highest reflectivity in the UV spectrum [Maninen and Taskinen, 2007]. Clear strong LSR peaks of ~0.15 sr-1
were discerned over Mongolia in winter (Fig. 14b), which are not present over Sahara (Fig. 14a), where LSR remains below
0.05 sr$^{-1}$. Moreover, these lower LSR values, compared to snow-affected months are typical for semi-arid regions like
Mongolia in spring and autumn (see September, October, April and May in Fig. 14b).

For other regions, the agreement between LSR and LER varies depending on the ecosystem type. For instance, in
ever-green ecosystems of southern hemisphere like tropical regions, the agreement is rather low or lacking. To be specific,
for southern hemisphere while LER intercomparison agreement is high for tropical forests like Southern-Hemisphere
Amazon region (r = 0.93), Indochinese Peninsula (0.86) or mixed ecosystems like South-Central Africa (0.96), Aeolus does
not exhibit any statistical agreement with either of LER references in any of these regions (r < 0.10). The dynamic range of
LSR variability is very low and close to instrumental noise magnitude of Aeolus in evergreen regions, as indicated by Fig.
15d showing Amazon region. Perhaps, Aeolus LSR is less sensitive to green vegetation changes at lower reflectivity ranges
or has weaker returns from green surfaces, thereby reducing dynamic range of LSR change over such areas [Weiler et al.,
2021]. We should stress that this suggestion is merely a hypothesis, which necessitated deeper exploration, shown in the next
section. In the mixed vegetation regions, the agreement patterns may vary with the best LSR-LER agreement in Australia
(Fig. 15) with LSR agreeing well with both LER$_G$ and LER$_T$ (r = 0.50 and 0.94, respectively). Thus, seasonal surface
changes can be resolved using Aeolus LSR even in the regions without snow cover. The presence of snow cover over the
region called Australia in this paper is unlikely because we focused solely on central Australia, excluding mountainous
regions where snow pixels might be present. In ocean areas, like the Guinea Gulf depicted in Fig. 15f, either LSR-LER
agreement is low or there's a lack of data for comparison.






**Fig. 14** LSR (red and blue) comparison with LERT (black) and LERG (grey) for several representative regions including: a –
Scandinavia (Region with frequent snow occurrence e.g. "snowy region" as described on the plot), b – Mongolia (Semi-
Arid), c – Sahara (arid), d – South-Hemispheric (SH) part of Amazon (Evergreen region), e – Australia (Mixed Region), d –
Guinea Gulf (Ocean Region). Error bars are taken from one-sigma monthly deviations of average LSR. Red-colored and
blue-colored plots have different y-axis ranges (red – strongest LSR regions, blue – weak-to-moderate LSR regions).



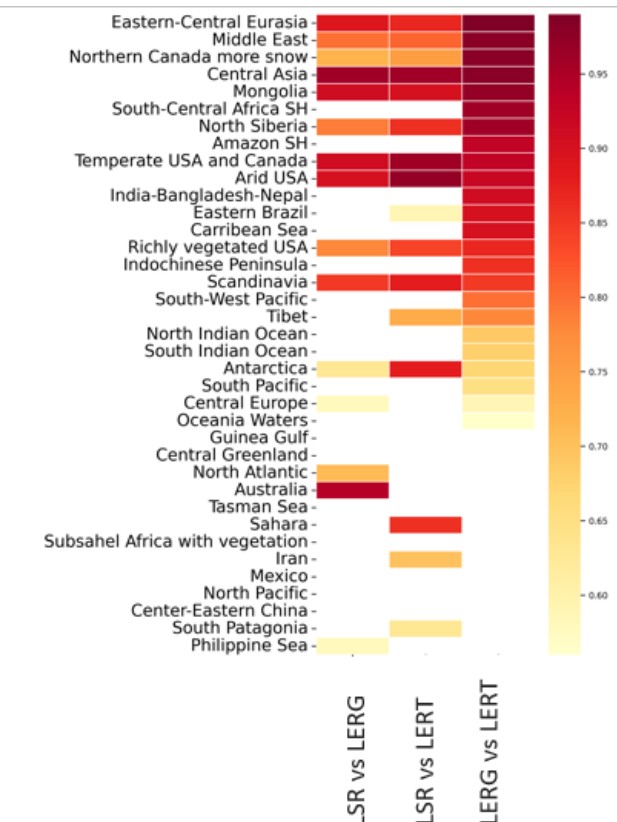

**Fig. 15** Correlation table between Aeolus (LSR), GOME-2 (LERG) and TROPOMI (LERT) monthly regional averages for all regions considered in this study. The correlation table is sorted in descending order of correlation coefficient between LERG and LERT

**3.4 Sensitivity of LSR to land cover: snow cover and vegetation proxy**

Thus far, we registered good linear agreement between LSR and LER references at orbit, aggregated monthly orbit, yearly global and regional monthly scales for most cases. Moreover, we have previously revealed a distinct clustering of annual LSR regional averages in Labzovskii et al. [2023]. This clustering closely reflected the ecosystem characteristics of different regions, with the LSR magnitude in the following ascending order: ocean regions, highly vegetated regions, arid regions, snow cover-prone region. On top of that, we had previously registered moderate negative correlations between yearly averaged LSR and NDVI values, which indicates a simple relationship – less vegetation, stronger LSR. However, all these results do not directly demonstrate the ability of Aeolus to resolve ecosystem-driven changes of land surface. To remind, we had arbitrarily selected the regions that reflect different geographical characteristics worldwide. To this end, we





quantitatively evaluated two previously suggested and the most promising hypotheses stemming from all our LSR works
[Labzovskii et al., 2021; 2022; 2023], namely; (1) strong sensitivity of Aeolus LSR to snow cover change and (2) moderate
sensitivity to vegetation change. To this end, we selected two proxy datasets for reflecting these characteristics including
snow cover and vegetation proxies – NDVI; both parameters were described and explained in the methodology.

Like in the aggregated orbit analysis, shown in Fig-s 11 and 12, we sampled the modelled values reflecting land
cover conditions to every clear LSR observation for every month. The analysis of snow cover (Fig. 16) unveiled very high
agreement with LSR yielding positive correlation for all months during the analysis. The highest agreement was found for
November and December 2018 (r = 0.74 for both months) seemingly due to the high dynamic range of snow cover during
these months and high differences between snow-covered high latitudes of NH and other regions in these months. In other
months, the correlation was moderate (r = 0.60, 0.62 and 0.68 for February 2019, March 2019, September 2018) or high as
well (r = 0.70, 0.71 and 0.72 for April 2019, January 2019 and October 2018). Regarding vegetation, we evaluated the LSR-
NDVI hypothesis by estimating statistical agreement between LSR and vegetation cover (see Fig. 17). We found weak-to-
moderate negative association between LSR and the vegetation reference – NDVI with the strongest negative correlation in
November 2018, December 2018 and January 2019 (r = -0.62, 0.61 and 0.60, respectively). For other months, rather weak
negative association between NDVI and LSR was registered with correlations ranging from -0.48 (September 2018) to -0.59
(May 2019). We noticed an interesting pattern manifested by the highest agreement in the periods when snow cover is
highest in northern hemisphere. Moreover, during these months, two distinct populations of LSR with negative association:
stronger LSR (see the horizontally prolonged upper population) and the lower LSR population of the same shape. The
stronger LSR population is distributed across the entire NDVI range, while the lower shape population mostly ranges from
0.5 to 1.0. Since the stronger population density on the plot is lowest in September and the agreement with the snow cover is
also lowest in September, we suspect that the stronger LSR population is related to snow cover occurrence, not vegetation
directly. This suggestion is sensible since numerous previous studies, many of which were mentioned by Taskanen and
Manninen [2007] had demonstrated that UV reflectivity of green Earth surfaces is very weak, unless covered by a white
layer such as snow [Warren 1980, Wiscombe 1980, Chyleck 1983, Grenfell 1994, Feister 1995, McKenzie 1996, Frei 1999,
Robinson 1999, Wuttke 2006, Weiler 2017]. To evaluate this hypothesis, we applied a snow cover mask of 0.05 and filtered
out all observations above this threshold. This evaluation indirectly confirmed our suggestion as shown by supplementary
Fig. E4. Once we filtered out all snowy cases, the stronger LSR population at the higher segment of the plot nearly
disappears (see Fig. S8 in the supplementary). Most crucially, the negative correlation between NDVI and LSR dwindles to
very low values (r < 0.30) for every month of the analysis if all snow cover cases are masked out.



**Fig. 16** Monthly scatterplots comparing LSR (sr-1) and snow cover (SNW_1) during entire FM-A period. Y-axis and
colorbar are both shown in logarithmic scales.





**Fig. 17** Monthly scatterplots comparing LSR (sr-1) and NDVI (VEG_1) during entire FM-A period. Y-axis and colourbar
are both shown in logarithmic scales.



To finalize about the discussion on the sensitivity of LSR to NDVI, we also evaluated the bin-based agreement of LSR versus snow cover, LSR versus NDVI, LSR versus NDVI without snow (snow cover < 0.05). As seen from Fig. 18 below, the agreement of LSR with NDVI (grey colour markers) is mostly driven by the changing snow cover (e.g. more snow, lower NDVI). The snow cover (blue colour markers) is in nearly ideal antiphase with NDVI while both compared with LSR. The pattern here is as follows – the higher snow cover, the higher LSR. At the same time, another side of this pattern is – the

lower NDVI, the higher LSR as well. However, if we filter out all snowy cases from NDVI (see red color markers on Fig. 18), LSR remains nearly unchanged and weak (below 0.05 sr$^{-1}$) across all variations of NDVI. It is unclear why NDVI binning does not reflect the pattern we noticed in previous paper namely a distinct gradient between rich vegetation and arid regions, as well as moderate negative agreement between yearly averaged NDVI values and yearly averaged LSR values at regional lever [Labzovskii et al., 2023]. Perhaps, the LSR difference between arid and vegetated regions is lower than we

expected prior to this work and LSR is mostly sensitive to the appearance of white surfaces [Taskanen and Manninen, 2007]. We discuss another suggestion explaining this phenomenon in discussion in detail.





**Fig. 18** Bin-based plots (50 averaging bins), illustrating: NDVI with snow (VEG-Y), NDVI without snow (VEG-N) and snow cover cases (SNW) for every month in the FM-A period.


## Discussion

We described the methodology behind the Aeolus lidar surface return (LSR) retrieval algorithm to be incorporated as
the official Aeolus Level 2A product during the post-commissioning phase. In short, the algorithm relies on the combination



of Aeolus L1B (information about ground bin detection and geolocation parameters) and L2A AEL-PRO data (backscattering coefficient) for calculating the LSR as surface integrated attenuated backscatter for all bins, where ground return was located using the official surface detection algorithm. The ability of Aeolus to resolve optical characteristics of Rayleigh contribution, aerosols and clouds made the atmospheric correction procedures simple and effective. We (1) used

Rayleigh and aerosol optical depths calculated from L2A molecular backscattering and aerosol extinction of AEL-PRO data, respectively, as well as (2) quality flagging of LSR signal. To include only useful LSR observations for the analysis, one should account for the number of attenuative features over ground bin and aerosol conditions, both can be estimated from the atmospheric quality flag ('qflag') we introduced and AOD, calculated using extinction from Aeolus L2A data. As a minimum quality assurance procedure, we strongly advise to include only clear LSR observations, namely, only those

observations satisfying qflag = 0 (no attenuative features above ground bin) and AOD < 1.0 conditions.

According to the official algorithm of ground bin detection, of all the Aeolus soundings in FM-A period, the ground bin was detected in 8 – 22% cases per month (19% cases in median) and clear useful LSR observations were available in 7 – 16% cases per month (14% in median), depending on month. The largest number of clear LSR observations were available from November (2018) to April (2019), seemingly due to the presence of strongly reflecting white surfaces in the northern

hemisphere. Importantly, the LSR algorithm was shown to be relatively stable to the change of AOD threshold (0.5 – 1.5), therefore indicating its potential for being used as Level 3 like gridded product at the Aeolus given observational data abundance. The official Aeolus ground detection algorithm yielded fewer ocean surface returns due to the weakness of water signal, compared to our previous work [Labzovskii et al., 2023]. Other LSR differences with the aforementioned work were minor and were simply driven by different data filtering strategies and which observations are deemed to have clear LSR.

Since land and ocean LSR demonstrate not only different magnitude of return in terms of signal, but likely different physical effect in returns, our results prompt us to create another holistic quality flag for LSR (or hflag) for users. In this context, 'hflag' can reflect three conditions including type of surface (0 – water, 1 – land), presence of cloud-driven attenuation over the ground bin (0 – no attenuation, 1 – more than one attenuative feature is detected and LSR can be therefore noisy or not representative) and presence of aerosol-driven attenuation over the ground bin (0 – low aerosol load, no attenuation, 1 –

potential aerosol attenuation). Users are advised to use the '000' flag for land surface reflectivity-oriented studies.

The detailed examination of Aeolus LSR during the FM-A period conducted in this study unveiled interesting results. Monthly average gridded LSR forms distinct clusters and varies from very weak returns of < 0.0018 sr$^{-1}$ registered over water surfaces to the range of 0.018 – 0.080 sr$^{-1}$ typical for land surfaces without snow, up to 0.080 < LSR < 0.417 sr$^{-1}$ values emerging in regions with occasional or permanent snow/ice cover. Such LSR signal distribution makes Aeolus non-nadir UV

reflectivity pattern very different from CALIPSO near-nadir visible reflectivity pattern. The CALIPSO LSR pattern previously exhibited strongest reflectivity returns from deserts and ocean surfaces and did not exhibit any exceptionally weak returns, compared to land [Lu et al., 2018]. In our work, the brightest sea ice returns are the highest being ~26 times stronger than the strongest water returns, resembling the magnitudes of UV returns from the same type of surfaces from Chadysienne [2008]. Unlike the Labzovskii et al. [2023] work, this detailed study revealed no differences between arid and



vegetation regions, but we noticed a previously unseen gradient between snow (in most cases < 0.160 sr$^{-1}$) and sea ice in Antarctica or Arctic waters (0.160 – 0.417 sr$^{-1}$) on global 2.5° x 2.5° LSR maps. In terms of LSR evaluation, we achieved a very good agreement between LSR and LER references (both GOME-2 and TROPOMI) at nearly all spatio-temporal levels. Four reference orbits we selected from 2018.09.10, 2018.11.30, 2019.01.11 and 2019.05.01; all exhibited reasonable agreements in terms of LSR-LER comparisons. Correlation coefficients ranged from 0.55 to 0.71 in Aeolus-TROPOMI and

from 0.57 to 0.65 in Aeolus-GOME-2 comparisons, respectively, whereas the agreement was mostly driven by land LSR. For monthly aggregated orbits, containing all clear LSR observations and corresponding sampled LER values from climatologies, we found moderate-to-good agreement for Aeolus-TROPOMI (ranging from r = 0.61 in February 2019 to 0.77 in September 2018) and weak-to-moderate agreement in Aeolus-GOME-2 comparisons (ranging from r = 0.44 in February 2019 to 0.64 in September 2018). The absence of perfect linear agreement is attributed to the distinct physical

behaviors of LSR and LER, which vary depending on surface changes. Unlike quasi-linear growth of LER, LSR exhibits a sigmoid-like increase in reflectivity when transitioning from a dark to a white surface. At regional level, seasonal dynamics of LSR agreed very well with LER dynamics in snowy regions (North Siberia, North Canada, Eastern-Central Eurasia, Scandinavia; r > 0.90), arid regions (Sahara, Middle East, Iran; r = 0.70 – 0.86), semi-arid regions (Mongolia, Central Asia, Western U.S., r = 0.90 – 0.97) and some regions with mixed vegetation as Australia (r = 0.94 in Aeolus-TROPOMI

comparison). However, in greener regions, the agreement between seasonal dynamics of LSR and LER is lower or non-existent due to low dynamic range of reflectivity and weaker sensitivity of Aeolus LSR to green surfaces we discuss below. At the global level, averaged 2.5° x 2.5° LSR estimates for the entire FM-A period exhibited excellent agreement with the averaged LER estimates yielding correlations of 0.90 with GOME-2 and 0.92 with TROPOMI, respectively.

The expectations that Aeolus LSR is extremely sensitive to snow cover changes were confirmed in this study. On the

aggregated monthly orbit level, we found a very high agreement between modelled snow cover and LSR with correlation ranging from 0.62 in March 2019 to 0.74 in November and December 2018. These results directly confirm both literature-based expectations about exceptionally strong reflectivity of white surfaces at UV for lidars [Weiler, 2017] and our previous suggestion about sensitivity of Aeolus LSR at UV to snow cover. For NDVI, we found some complex interdependencies between vegetation cover and snow cover. Due to a nearly ideal antiphase of snow cover with NDVI while both compared

with LSR, we decided to also filter out all snowy cases from NDVI-LSR comparison as an additional examination. After such filtering, nearly no change of LSR depending on the NDVI change could be registered and LSR remained fairly weak with values mostly below 0.05 sr$^{-1}$. Possibly, Aeolus exhibits less sensitivity to variations among different vegetated surfaces. Alternatively, in moderate latitudes, where most Aeolus clear LSR observations are available, the vegetation frequently stretches upward past the snow layer, substantially affecting the snow-covered terrain's reflective capacity

depending on land cover type and snow condition [Taskanen and Manninen 2007]. The mechanisms tied to the transition of surface albedo in vegetated areas with occasional presence of snow can therefore manifest a very complex interplay that is challenging to disentangle in the current paper.



## Conclusions

While the main aim of this study was to familiarize readers with the Aeolus Lidar Surface Return (LSR) algorithm and resultant LSR parameter, we also showed that gridded LSR estimates manifest some reflectivity and land-cover-relevant patterns can be useful for researchers. We encourage a production of not only operational LSR data at the original resolution but also a gridded 2.5 x 2.5° LSR product in the form of Aeolus-lifetime LSR Level 3 climatology as a future effort. Leveraging Aeolus LSR's excellent sensitivity to white surfaces like snow cover, particularly in high latitudes, such products

can significantly benefit researchers interested in radiative transfer, reflectivity, and snow cover studies. Most crucially, together with CALIPSO-based works on land surface returns [Lu et al., 2018] and ocean surface returns [Josset et al., 2008; Hu et al., 2008; He et al., 2016; Venkata and Reagan, 2016], our methodological framework and scientific results on Aeolus LSR will complement Aeolus post-commissioning studies and pave the way for future lidar missions. These efforts have partially addressed the gaps in understanding surface LSR at UV and non-nadir angles by Aeolus, thereby deepening the

existing understanding provided by CALIPSO studies mentioned above. Considering these experiences collectively, a practical framework on maximizing the benefits from surface return signals for future lidar missions like EarthCARE and even Aeolus-2 can now be outlined. EarthCARE, with its operation of collocated radar and UV nadir lidar measurements, suggests that lidar surface returns can be utilized to construct UV surface climatologies sensitive to white surfaces. Additionally, EarthCARE's ocean surface returns will aid in implementing Aerosol Optical Depth (AOD) algorithms based

on both the inverse relationship of ocean surface backscatter and wind speed [Hu et al., 2008], as well as combining lidar with radar surface returns to infer AOD [Josset et al., 2008]. For Aeolus-2, assuming a similar optical setup, it is clear that AOD retrieval using ocean surface returns will be challenging, but efforts to deliver subsurface reflectance contributions of the ocean and continuation of the LSR 2.5 x 2.5 gridded record, which will build upon current efforts, would be highly beneficial. Moreover, the current LSR efforts can be used for modelling expected reflectance from Aeolus-2 in lidar

simulation tools.

**Competing Interests:** At least one of the (co-)authors is a member of the editorial board of Atmospheric Measurement Techniques.

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
