# Peer review of "Aeolus Lidar Surface Returns (LSR) at 355 nm as a new Aeolus L2A product"

_EGUsphere, 2024_

## Referee Comment (RC2)

**General comments** (initial paragraph evaluating the overall quality of the preprint):

This manuscript is a relevant piece of work paving the way for an effective design and usage of LSR in future lidar missions. It contributes to the expansion of the database for the research field of surface reflection properties from only passive to also active instruments. The results on the excellent sensitivity of Aeolus LSR's to white surfaces (snow and ice) are very promising for future studies. The manuscript addresses relevant scientific questions and presents a novel concept and data resulting in substantial and reasonable conclusions. The objectives of the paper are well described at the end of chapter 1. The method is described to a sufficient extent, so that reproduction by fellow scientists seems feasible. The structure of the overall presentation is reasonable, but the language might need some revision in certain paragraphs. Own new contributions are indicated and proper credit to related work is provided including a good number of references of high quality. The amount and quality of the supplementary material seems appropriate.

I recommend the acceptance of this manuscript after the necessary revisions. Specific minor comments and unfortunately a large number of technical corrections are suggested below.

**Specific comments** (addressing individual scientific questions/issues):

1. Line 1: Why is "Phase-F" mentioned so prominently in to paper title? Isn't "new" enough information? As "Phase-F" appears only one more time in the whole manuscript, I recommend to delete this term from the title.
2. Line 10: The term "incidence angle" might need additional reference information (incident onto what?). Maybe consider using the term "off-nadir" angle. (Further cases in lines 67, 97, …)
3. Line 22: Why are the reference orbits only from the period where Aeolus used its laser A? What about data from laser B with L1B 7.12? Please mention this including the used baseline version. → See also line 107 and Table 1. Are different results or performances expected from FM-B data?
4. Line 36: It might be worth noting shortly here that Calipso measured at a different wavelength than Aeolus.
5. Line 67: What do you refer to here specifically with "unique Aeolus setup"? In which way does it constitute a challenge to retrieve robust LSR. → Presumably you mean the coarse vertical resolution.
6. Line 120: Table 1. As you introduce the "reprocessing product" in line 107, it would be good if you add the Baseline to the column "Version" at least for the L1B.
7. Line 139: Could you provide a reference that shows how real surfaces can behave in terms of angular dependent reflection?
8. Line 149: Why do you perform a resampling from 0.25° to a coarser resolution of 2.5°?
9. Line 165: Why can't NDVI become negative (see equation in line 164)? Is the absolute value of NIR always equal to or larger than that of VIS?
10. Line 187: To my knowledge the Aeolus mission uses the ACE-2 DEM: https://sedac.ciesin.columbia.edu/data/set/dedc-ace-v2.

11. Line 189 ff.: To my knowledge the Aeolus ground detection looks for signal drops going upwards (in terms of altitude) first and then looks for signal drops going downwards.
12. Line 200: Northern or southern hemisphere winter in general, or both? Or do you refer to cases b and c?
13. Line 226: Assuming that the ground signal (disregarding the atmospheric contribution to the signal of the bin) emanates from a more or less infinitesimally thin layer, why shall it make sense to multiply here with the thickness of the whole bins (apart from getting a useful unit)? Doesn't the variation of the range-bin-thickness (0.25 km – 2 km, line 97) inadequately affect the LSR'?
14. Line 365: Can sea ice already be expected close to these latitude thresholds, e.g. around +/-40°?
15. Line 367: What should be the reason for a low amount of sea ice right after the southern hemisphere winter?
16. Line 429: Do you have any idea what could be the reason for the ocean returns being place exclusively in such regions? Algae (might explain the seasonality), white caps (maybe rather the cyan values around -60° latitude?), microplastics, …? It might be worth putting the info from line 474/476 already here?
17. Line 440: Figure 6, top left panel: Could you comment on the apparent increase of valid data in Africa (Sahel zone, region south of African rainforest, Mozambique), South America (north and south of the rainforest) and South East Asia from September to November? Is it related to the general unavailability of data, or could it have a meteorological (or another) reason, such as the movement of the ITCZ or cloud climatology? Or are lines 453-457 sufficient for explanation?
18. Line 460: Did you encounter a decrease in the number valid LSR data over the FM-A period correlated with the decrease of the laser energy of Aeolus?
19. Line 466/467: To me this doesn't look localized, but rather like a clear trend for the whole arctic region for these three months, as also Greenland shows this behaviour. But can a potential "wettening" of the ice/snow with warmer temperatures be consistent with increasing LSR? By melting snow leaving behind the below ice surface?
20. Line 490: It would be good to have this rough classification and allocation of LSR values to surface types already before Fig. 6.
21. Line 498/499: Why only the Southern Ocean? I see a lack of valid signals over all ocean areas between +/-70° lat.
22. Line 504: I could well imagine here a 4-by-4 plot with maps of the number of valid Aeolus observations per grid point, in order to get an idea of the distribution and significance of the comparison for certain regions.
23. Line 524: What's the reason for showing the y-axes in log scale, unlike for Fig. 11+12?
24. Line 814: I would recommend to also explicitly mention the excellent sensitivity of Aeolus LSR to white surfaces (snow/ice conditions) in your abstract.

**Technical corrections** (typing errors, etc.):

1. Line 12: Although well known, the first appearance of the abbreviation "UV" could be written out.

2. Line 24: Assure consistency for LERG and LERT naming with (l. 137 ff, l. 518 ff., 540 ff., 572, 604 ff.) and without subscripts.
3. Line 33: The "we demonstrated" seems to be either a left over or misplaced (parentheses?) to me.
4. Line 49: … at the lidar …
5. Line 100: I assume the either "both" or "two" is redundant.
6. Line 111: As you explain in line 114 that FM stands for feature mask, you might also want to state what PRO stands for.
7. Line 119: described
8. Line 150/151: no comma needed between "estimates" and "reflected".
9. Line 152/153: Because of the use of "although" this seems to be an unfinished sentence.
10. Line 176: … where the DEM …
11. Line 179: … from the ground …
12. Line 182: Comma after "Aeolus bin" really necessary?
13. Line 188: Is "sought" correctly used here? Alternatively: "searched", "looked for" or "sought for" (also line 176).
14. Line 189: … below the height of the DEM …
15. Line 189:  Proposal: "The algorithms searches for the highest bin that contributes to the ground signal, starting from the first bin with non-negative valid useful signal."
16. Line 191: Proposal: "… in upward direction."
17. Line 191: lowermost
18. Line 199: What shall be expressed by the " - " between "potentially" and "noise"?
19. Line 222: … and the highest …
20. Line 226: Eq.1 should most probably start with " LSR' " instead of " y,". Otherwise introduce y' in the text here, as done with y before Eq.4.
21. Line 226: Eq.1 – missing explanation of $z_i$
22. Line 244: … the optical depth we …
23. Line 250: overestimation
24. Line 250: probably two spaces after "situated"
25. Line 255: … the given ??? …
26. Line 263: aerosol
27. Line 266: two commata after "(HSRL)"
28. Line 270: RayOD → $OD_{Ray}$
29. Line 271: deleted the second "both" in the same sentence
30. Line 280: … of the LSR …
31. Line 281: Please explain/write out AB already here than in line 329.
32. Line 295: probably two spaces after "echo"
33. Line 318: … calculating the scientific …
34. Line 320: probably two spaces after "year"
35. Line 324: Fig.2 contains "L1A" in the top box, although this has not yet been mentioned. Perhaps you mean L1B.
36. Line 330: orbits
37. Line 330: Proposal: "… with colored indexed for the selected observations placed in each subplot."
38. Line 334: … or weakened the LSR …
39. Line 336: qflag

40. Line 340: … having the highest …
41. Line 348: … ground bins …
42. Line 352: Fig.3 could have a more expressive design, e.g. with filled, colored and semi-transparent background boxes.
43. Line 353: (ATB) should be deleted here, because it defines a second acronym for a quantity that has already been defined before.
44. Line 354: According to the text above case 1 (red frame) has a qflag = 0%.
45. Line 357: Proposal: "Note that the term *index* above each subplot stands for the respective orbit number of the Aeolus mission."
46. Line 363: This is not a full sentence.
47. Line 383: Would be nice to have all four plots with the same x-axis range.
48. Line 384: latitude
49. Line 370/371: There is no error propagation mentioned in the caption of Figure 4.
50. Line 374: Within the parentheses the values for water seem to be missing.
51. Line 374: … land are explained …
52. Line 377: … are clipped for water surface from …
53. Line 388: covers
54. Line 392: probably two spaces after "hypotheses" and before "strong"
55. Line 398: probably two spaces after "we"
56. Line 399: probably two spaces after "Several"
57. Line 440: The caption mentions panels a-f, but the letters do not appear anywhere.
58. Line 429 (same for line 460+477+503): The title textboxes of the lower four plot seem to overlap with their above plots. Resolution of the figure could be improved. What about the information on the top left? Lat/Lon info for the y/x axes would be helpful.
59. Line 490: Proposal: … representing mostly sea ice … (red colour appears also elsewhere)
60. Line 533: Is this a second caption to the same table? Please unify the information.
61. Line 540: LER instead of LERs
62. Line 542 ff.: Please insert the values for the correlation coefficients also in Figures 11 and 12, as done for Fig. 13.
63. Line 547: … along the y-axis …
64. Line 548: Proposal: "various snow types and conditions" instead of just "snow"
65. Line 548: … by a strongly …
66. Line 553: … such an experimental …
67. Line 556: … is increased to r = …
68. Line 564: Fig. 11: Not a good resolution. Labels of y-axes overlap with colour scale of neighboured plots. Caption: LER should be LERG or LERG or $LER_G$. Please make consistent with x-axis labels.
69. Line 573: probably two spaces after "the"
70. Line 575: delete "comparisons"
71. Line 576: Does a second "agreement" make sense here?
72. Line 579: Fig. 12: Not a good resolution. Caption: LER should be LERT or LERT or $LER_T$. Please make consistent with x-axis labels.
73. Line 586: Fig. 13: Panel a and b are mentioned in the caption but are not noted anywhere in the figure. → use left and right?
74. Line 598: … from an ecosystem …
75. Line 599: … and a correlation …

76. Line 601: probably two spaces after "was"
77. Line 605: Fig 14c refers to the Sahara region, not to snow covered Scandinavia (a).
78. Line 607: Where do you see the ~0.3 of LER(G/T?) in Fig. 14a?
79. Line 608: "Northern Canada" … not shown here
80. Line 610: Where are these values for Central Greenland in Fig. 15? →
81. Line 611ff: Do these values match the colours in Fig. 15, or ist the colour scale wrong?
82. Line 614: probably two spaces after "changes"
83. Line 619/620: Grammatically incorrect first part of the sentence.
84. Line 624: very god agreement only too LERT but obviously not to LERG
85. Line 655: Fig. 14. You show LERT and LERG, but mention only LERG in the labels of the y-axes. What about the availability of error bars for LERT?
86. Line 625: Fig. 14a refers to Scandinavia but not to Sahara
87. Line 625: sensitive → sensitivity
88. Line 634: Fig. 14a refers to Scandinavia but not to Sahara
89. Line 638: … of the southern …
90. Line 643: 15 d → 14 d
91. Line 648: Refer to Figure 14 e.
92. Line 650: 15 f → 14 f
93. Line 714: Fig. 16 caption: … during the entire …
94. Line 714: Fig. 17 is clipped at the top. Caption: … during the entire … colorbar …
95. Line 695: This is not a fully sentence in a grammatical sense. Also "moreover" doesn't make sense here as you already referred to the pattern in the previous sentence.
96. Line 699: Fig 16 – 17 – 18: Bad resolution → hardly readable label of y-axes. Please use capital letters for "LSR" in the labels.
97. Line 703: Please explain the value 0.05 here. What does it mean and where do you get it from? Is it 5% of the area covered by snow?
98. Line 705: Fig. E4 → Fig. S8
99. Line 706: No need to refer to the same figure again.
100. Line 734: Fig. 18. Not well arranged with partly mutual clipping of the subplots. Does "snow cover cases" represent a count/number here? If so, it needs a second y-axis. In Figure 16 it is "snow cover" only. Otherwise formulate the sentence more clearly. Think of a more appropriate label for the x-axes or add a second one for SNW. Are the grey bars error bars?
101. Line 727: … in our previous paper …
102. Line 729: level
103. Line 731: … in the discussion chapter in …
104. Line 740: What do you mean with "during post-commissioning phase"? Doesn't the commissioning phase of a satellite comprise only the first few weeks to months? If so, this would be long ago for Aeolus, but you seemingly want to express that your LSR product is to be placed into the Level 2A product in the near future, which is already after the end of the Aeolus mission.
105. Line 751: … in the FM-A …
106. Line 760: … different magnitudes of return signal … effects in the returns …
107. Line 768: Decide for one consistent representation of the ranges, either "x – y" or "x < LSR < y"
108. Line 791: … to a low …
109. SUPPLEMENT Line 19: What does "grind" mean in the x-axis labels?

---

## Author Comment (AC1)

**RESPONSE TO REVIEWERS**

We sincerely thank the reviewers for the helpful comments that will improve the quality of our manuscript. We incorporated these suggestions and responded to the inquiries which required clarification or argument-based opinion from our side. Please see our responses below in regular font, while reviewers' comments are shown in bold.

**RESPONSE TO REVIEWER#1**

**1. The methodology in this manuscript should be well-constructed, it should work fine in this study. My main concern is the framework of this manuscript, regarding the sequence of section 3.1 and section 3.2. Normally, one should firstly verify the method proposed in this paper before the statistical analysis. I mean, probably authors may think about describing section 3.2 before section 3.1.**

Response 1.1: We agree with you on this remark. We have now revised the manuscript to present section 3.2 before section 3.1, as you suggested. Additionally, we would like to highlight that the method has already been validated on a limited scale in our previous work (Labzovskii et al., 2023; https://www.nature.com/articles/s41598-023-44525-5), which provides a foundational basis for its more extensive examination in the current study.

**2. For the sections of "Discussion" and "Conclusion," I propose the authors to reorganize these two parts. Some information in the Discussion part**

**should be moved to the "Conclusion" part. In the current manuscript, the conclusion somehow looks like an outlook.**

Response 1.2: Often, researchers read only introduction and conclusions, making this section essential to inform them about key takeaways. In this light, we believe that well-supported conclusions provide an excellent opportunity to offer a perspective of researchers on the scientific findings in either the form of summary or outlook. We opted to frame this perspective as an outlook because the Aeolus mission was the first to apply this methodology to UV LSR. Since Aeolus has been decommissioned we decided to provide in conclusions Aeolus-based guidance and recommendations for future and upcoming satellites. This decision aims to assist researchers interested in implementing such algorithms but who may not be fully aware of the exact opportunities and challenges of highly non-nadir UV LSR. Meanwhile, our discussion summarizes abundant results this study provided in a convenient, short way. I hope the current structure, being supported by such rationale is acceptable.

**Technical corrections:**

**3. The symbols of variables in the article should be consistent. For example, qflag (sometimes quality flag)/LERG (sometimes LERG)/LSR (sometimes lsr)/ODray (sometimes RayOD). Please make them consistent.**

Response 1.3: We standardized all annotations here. Full name TROPOMI LER (when no space – LERT), full name GOME-2 LER (when no space – LERG; full name quality flag (when no space – qflag); RayOD was a typo. All inconsistencies and subscripts addressed.

**4. Line 317: Why 1-sigma but not 3-sigma is selected? Is 1-sigma too strict for the calculation?**

Response 1.4: The use of 1-sigma is appropriate in this context to ensure that we capture statistically significant variations while maintaining robust results. It is not uncommon to use 1 sigma as the uncertainty proxy for gridded atmospheric datasets in research (see Han et al., for example - https://essd.copernicus.org/articles/15/3147/2023/essd-15-3147-2023.html). Note that this is arbitrary choice that has not affected results in any way.

**5. Figure 3: Red frame: qflag > 0, but the QF in the attenuated case with number 1 is 0. Please check it.**

Response 1.5: The formulation on the figure caption was somehow misleading, despite being technically correct. We rephrased the statement in the middle of Fig. 2 for clarity like *'clear observations: AOD < 1.0; qflag = 0)'*. To clarify here, only the cases with qflag = 0 are considered clear and therefore, only such cases are used in the final analysis.

**6. Figure 4: I cannot see the histogram distributions with magenta color. In the text, of course, you explain only a few numbers of strong return remain; therefore, please find another way to show the distributions of water reflectivity between -35 and 35.**

Response 1.6: We have addressed the issue with Figure 4 and have updated the histogram distributions to improve visibility. The new presentation on Fig. 4 illustrates three distributions with log y-scale, whereas water returns between -35

and 35 latitude are shown in red, rather than transparent magenta, which also improves visual comprehension aspect.

**7. Actually, the authors should reproduce all the figures in this manuscript. They are hard to be recognized, especially for the legends/values.**

Response 1.7: We acknowledge that many figures were initially provided at a resolution lower than dpi 300, while having misalignment issues and caption ambiguities. All figures have been finalized; we have now replaced all figures with higher resolution (dpi 300) to enhance clarity. We have modified most figures to improve their readability, as specified below. For figures with multiple panels, we have made efforts to optimize the quality at 100% zoom. Except dpi enhancement and optimizing the font for 100% zoom at Microsoft Word (as required by the AMT formatting guidelines), we list specific changed below (note that we use the original figure numbering below from version 1 of the manuscript, before sections 3.1 and 3.2 were flipped)

- Figure 1 (all panels and captions aligned)
- Figure 2 (the font of central panel is enlarged; unclarity on the remarks about filtering procedure on '3' is addressed; the statement rephrased to *"Filtering out low-quality LSR (clear observations: AOD < 1.0; QFLAG = 0)"*)
- Figure 3 (only dpi and fonts changed)
- Figure 4 (For better visibility, we changed y-axis to log-scale visualization and changed magenta transparent color for denoting no-ice water to red

non-transparent color; other changes are cosmetic and were applied to improve the aesthetics of the figure)

- Figure 5 (observation points increased, panel annotations 'abcd' assigned, captions aligned, frames created)
- Figures 6 – 8 (replotted with higher dpi, panels were annotated 'abcd…' and aligned)
- Figure 10 (the size is significantly enlarged; every panel was annotated as a, b, c, d)
- Figures 11 – 13; 16 – 18 (Panels annotated as 'abcd…', panels aligned, titles added, captions corrected; z-axis color annotated)
- Figure 14 (panels annotated, region annotations aligned, y-axis changed to "LER" to avoid confusion)
- Figure 15 (the font size of correlation caption increased, the figure size increased as well)

**8. Figure 10: Is it necessary to use log scale for LSR?**

Response 1.8: The use of a logarithmic scale for LSR is advisable due to the exponential nature of the LSR signal, which changes significantly from water to snow surfaces. Its behavior is in this aspect different from LER that changes more linearly and uniformly as the surface becoming whiter. In sort, the logarithmic representation effectively captures the wide range of signal variations. We have documented this phenomenon in our previous study (Labzovskii et al., 2023).

**9. Figure 13: Why only "LER snow TROPOMI" and "LER CLEAR GOME-2" are compared? The authors should also explain why LER with other features are not compared.**

Response 1.9: This rationale stems from our previous study (Labzovskii et al., 2023; https://doi.org/10.1038/s41598-023-44525-5) and was previously described. To quote the Data section from Labzovskii et al. 2023: *We used LER estimates, reflecting snow-affected areas as well because it is crucial to include snow/ice regions in the analysis to evaluate the sensitivity of Aeolus LSR to the strongest white reflectors*. Note that for GOME-2 this is the only available type of dataset at 354 nm we compared with. A more detailed explanation is given in 2.1.2 of the current manuscript.

**10. Figure 14: Which y-label is designed for TROPOMI? Is the y-label of "GOME-2 LSR" shared for TROPOMI and GOME-2? Please make it clear.**

Response 1.10: The caption here was misleading. Y-axis reflects LER for both GOME-2 and TROPOMI (despite being not identical, they are very similar in average terms and let alone, similar by magnitude). We have corrected the y-axis to 'LER' in this Figure 14.

**11. Line 680: How the values of snow cover (SNW_1) are calculated? Is it a proportion?**

Response 1.11: Snow cover information is taken from MERRA-2; an atmospheric reanalysis system from the NASA GMAO efforts (Gelaro et al. https://journals.ametsoc.org/view/journals/clim/30/14/jcli-d-16-0758.1.xml).

MERRA-2 uses the GEOS-5 model to simulate atmospheric processes like radiation and convection on a grid. It employs Gridpoint Statistical Interpolation (GSI) to integrate observational data, adjusting the model's outputs to reflect real-world conditions every six hours. In terms of sea ice, MERRA-2 assimilates recalibrated versions of some of the satellite observation types like for sea ice observations, primarily obtained from satellite passive microwave sensors, provide data on sea ice concentration and extent. We added this rephrased remark from GEOVANNI website description of snow/ice cover extent: "*...where the extent of snow or ice cover on the Earth's surface is used (the fractional amount of a land surface covered with snow and ice within a tile, ranging from 0 to 1).*"

**12. Figure 17: Please recheck this figure. The titles (month information) of the graph in the first row have been obscured and are not displayed entirely.**

Response 1.12: We have revised all figures, see Response 1.7 please.

**13. Line 738: Discussion should be listed as section 4.**

Response 1.13: Discussion is now labelled as section 4.

**14. Line 809: Conclusions should be listed as section 5.**

Response 1.14: Conclusions are now labelled as Section 5.

**RESPONSE TO REVIEWER #2**

**Specific comments (addressing individual scientific questions/issues):**

**Line 1: Why is "Phase-F" mentioned so prominently in the paper title? Isn't "new" enough information? As "Phase-F" appears only one more time in the whole manuscript, I recommend deleting this term from the title.**

Response 2.1: We deleted this from the title.

**Line 10: The term "incidence angle" might need additional reference information (incident onto what?). Maybe consider using the term "off-nadir" angle. (Further cases in lines 67, 97, …)**

Response 2.2: We provided the definition of incidence angle at the first occurrence in the introduction, to quote the introduction "*incidence angle of satellite instrument is the angle between the satellite sensor and the normal to the surface of the target cell.*" We kept the incidence angle in the abstract without explanation which is not uncommon for lidar studies. Namely, Kassalainen et al., 2018 is one among many examples we can provide - https://royalsocietypublishing.org/doi/full/10.1098/rsfs.2017.0033.

**Line 22: Why are the reference orbits only from the period where Aeolus used its laser A? What about data from laser B with L1B 7.12? Please mention this including the used baseline version. See also line 107 and Table 1. Are different results or performances expected from FM-B data?**

Response 2.3: Regarding the baseline version, we used the #3 reprocessing results in baseline 14 for FM-A period. We incorporated this remark in the methodological description where you pointed out. We used only FM-A period in this paper, from the reprocessing perspective, we have data input dependence with not only Aeolus L1 and L2A data, but also AEL-PRO algorithm alongside AEL-FM mask within. At the moment of the manuscript submission, we could rely only on the input from the FM-A period (before June 2019) in data

dependency. For the reprocessing #2 (FM-B) baseline 11 was used. These products did not contain the necessary variables to run the AEL-PRO (prototype) algorithm. Reprocessing #4 (FM-B part) was not yet released during this work so could not be used. Moreover, thus far, we have officially produced LSR datasets only for FM-A period using the #3 reprocessing and baseline 14 only, available for open access sharing but the FM-B period datasets should be produced soon from our side and we'll analyze this aspect independently. Some differences between FM-A and FM-B LSR outputs will be plausible according to our previous experience.

**Line 36: It might be worth noting shortly here that Calipso measured at a different wavelength than Aeolus.**

Response 2.4: We included this remark.

*"By taking together ==nadir-looking CALIPSO and highly non-nadir Aeolus experiences,== a framework on effective LSR utilization using future lidar missions such as EarthCARE and Aeolus-2 can be effectively designed."*

**Line 67: What do you refer to here specifically with "unique Aeolus setup"? In which way does it constitute a challenge to retrieve robust LSR? Presumably you mean the coarse vertical resolution.**

Response 2.5: We literally meant the highly-non nadir incidence of ~35° and 355 nm wavelength. This combination is unique itself for atmospheric lidar research regardless vertical resolution or other characteristics that may differ from CALIPSO for example.

**Line 120: Table 1. As you introduce the "reprocessing product" in line 107, it would be good if you add the Baseline to the column "Version" at least for the L1B.**

Response 2.6: The remark on the reprocessing # and baseline added in the paragraph over Table 1.

**Line 139: Could you provide a reference that shows how real surfaces can behave in terms of angular dependent reflection?**

Response 2.7: We added a reference from Maignan et al., (https://www.sciencedirect.com/science/article/pii/S0034425703003808?casa_token=cdF2rk0M74YAAAAA:JK-mZi4v_9D7788JC6KoHfj61sy0Hm0-2noc7fQFxkemMCn8TMFoIq_Rfr6Ru946M4moD_zC) study, who elucidated, to quote how to …"*to measure the bidirectional reflectance of a large variety of Earth targets*". For 22 000 sets of measured targets and evaluated reflectance models using real observations.

**Line 149: Why do you perform a resampling from 0.25° to a coarser resolution of 2.5°?**

Response 2.8: Because at 0.25°, there is an insufficient amount of Aeolus data to fill the grid cell. Therefore, grid comparison between LSR and LER should be done at a realistic resolution like 2.5°. This decision was made based on our previous experience from Labzovskii et al. (2023) that set our expectations for the balance between data abundance and the realistic resolution, of potential LSR L3 gridded product.

**Line 165: Why can't NDVI become negative (see equation in line 164)? Is the absolute value of NIR always equal to or larger than that of VIS?**

Response 2.9: In short, yes, for most vegetated areas, it cannot be negative because healthy vegetation reflects more NIR light than red light, making the difference between NIR and Red positive. Negative NDVI can occur over water or urban areas for example; both irrelevant to our study. We added a remark on this

"Negative NDVI values occur in scenarios where the reflectance properties are not typical of vegetation, like water, but such areas are outside of the scope of our analysis."

**Line 187: To my knowledge, the Aeolus mission uses the ACE-2 DEM: https://sedac.ciesin.columbia.edu/data/set/dedc-ace-v2.**

Response 2.10: We corrected the description to ACE v.2 DEM.

**Line 189 ff.: To my knowledge, the Aeolus ground detection looks for signal drops going upwards (in terms of altitude) first and then looks for signal drops going downwards.**

Response 2.11: Perhaps, our description was too vague. Please see if our current description incorporating your simpler, plainer explanation is better

"The height of the surface of the Earth with regard to the reference ellipsoid is used. Subsequently, the lower edge of each altitude bin is being sought, where the height of the bin should be below height of DEM. In short, the Aeolus ground detection looks for signal drops going upwards (in terms of altitude) first and then looks for signal drops going downwards. If ground bin candidates are more

*than five, the ground detection is not successful and therefore no ground bin is assigned for the respective observations [Lux et al., 2018]."*

**Line 200: Northern or southern hemisphere winter in general, or both? Or do you refer to cases b and c?**

Response 2.12: We rephrased that we did refer to the northern hemisphere here.

*"According to our previous experience [Labzovskii et al. 2023], more detected cases ==in winter in northern hemisphere== are explained by the presence of sea ice over ocean. As we are interested in land surface signal, we applied the surface flag mask"*

**Line 226: Assuming that the ground signal (disregarding the atmospheric contribution to the signal of the bin) emanates from a more or less infinitesimally thin layer, why shall it make sense to multiply here with the thickness of the whole bins (apart from getting a useful unit)? Doesn't the variation of the range-bin-thickness (0.25 km – 2 km, line 97) inadequately affect the LSR'?**

Response 2.13:

**Line 365: Can sea ice already be expected close to these latitude thresholds, e.g., around +/-40°?**

Response 2.14: The result here can vary depending on the methodology applied and on the assumption. For your interest, we duplicated the figure with -40 to +40 latitude boundaries below. Since only miniscule differences between two

choices and respective distributions are seen, let us keep the original figure with -35 < lat < 35 boundaries.

[Figure]

**Line 367: What should be the reason for a low amount of sea ice right after the southern hemisphere winter?**

Response 2.15: There are some studies we examined. To name one, Holland, (2014;

https://agupubs.onlinelibrary.wiley.com/doi/epdf/10.1002/2014GL060172)

showed that *"the ice-albedo feedback causes instability during **spring and summer**, whereby ice cover perturbations grow"* (e.g. spring ice loss is heavy

due to perturbations). Other reasons are wind trends and temperature effects (reduction trend starts exactly after winter in southern hemisphere near Antarctica).

**Line 429: Do you have any idea what could be the reason for the ocean returns being placed exclusively in such regions? Algae (might explain the seasonality), white caps (maybe rather the cyan values around -60° latitude?), microplastics, …? It might be worth putting the info from line 474/476 already here?**

Response 2.16: It is a very difficult pattern to disentangle because the official Aeolus algorithm filters out what is labeled as weak signal. However, we have a working hypothesis based on the letter we are preparing on ocean returns (mentioned in response 2.14), where we evaluated also assumingly weak ocean surface returns. It is a complex interplay between chlorophyll concentration and ocean surface conditions. In short, Aeolus ocean LSR is strong if chlorophyll is low and ocean mixing is low (tropical and sub-tropical gyres) with ~0.2 mg/m³ of chlorophyll as a threshold. Otherwise, once chlorophyll grows above this threshold, complete attenuation of Aeolus UV signal occurs and the signal becomes too weak or statistically disappears. We are trying to prove the validity of this pattern in the aforementioned letter and, unfortunately, cannot focus on this aspect more or provide this hypothesis without additional analysis and figures in the current paper.

**Line 440: Figure 6, top left panel: Could you comment on the apparent increase of valid data in Africa (Sahel zone, region south of African rainforest, Mozambique), South America (north and south of the rainforest), and South East Asia from September to November? Is it related to the general unavailability of data, or could it have a meteorological (or another)**

**reason, such as the movement of the ITCZ or cloud climatology? Or are lines 453-457 sufficient for explanation?**

Response 2.17: It can be related to ITCZ and also to the seasonality of west Africa biomass burning. We internally discussed a lot on this phenomenon. However, we do not have numerical arguments to prove this except for the actual result of these phenomena—namely high AOD from Aeolus. Due to this, we restricted ourselves to the explanation in lines 453-457 that is based on numerical AOD findings we have. In other words, we do not do deeper into the explanations we cannot prove here.

**Line 460: Did you encounter a decrease in the number of valid LSR data over the FM-A period correlated with the decrease in the laser energy of Aeolus?**

Response 2.18: We suspected this in terms of correlation and abundance, but it was not the case. See below the statistics of the successful ground detections (GD, official algorithm), cloud-free detections with aerosols (second column) and clear LSR estimates for the final analysis (third column). The abundance is more driven by cloudiness around the world, while for the global reflectivity, starting from January, global signal starts decreasing towards September (absolute minima), so downward trend towards May is plausible for this kind of data.

| date | All | Ground Detected (GD) | GD & qflag = 0 | GD & qflag = 0 & AOD < 1 |
|---|---|---|---|---|
| 2018/09 | 5,480,310 | 465,044 | 393,218 | 366,529 |
| 2018/10 | 6,398,400 | 645,696 | 520,056 | 482,108 |
| 2018/11 | 5,937,690 | 1,115,751 | 874,464 | 808,765 |
| 2018/12 | 6,213,600 | 1,255,028 | 972,504 | 898,912 |
| 2019/01 | 2,760,390 | 605,257 | 470,673 | 436,531 |

| | | | |
|---|---|---|---|
| 2019/02 | 2,659,800 | 520,681 | 359,191 | 330,899 |
| 2019/03 | 6,512,100 | 1,324,018 | 1,128,701 | 1,026,869 |
| 2019/04 | 6,263,430 | 1,168,668 | 949,454 | 859,650 |
| 2019/05 | 6,507,330 | 959,369 | 728,891 | 660,989 |

**Line 466/467: To me, this doesn't look localized, but rather like a clear trend for the whole Arctic region for these three months, as also Greenland shows this behavior. But can a potential "wetting" of the ice/snow with warmer temperatures be consistent with increasing LSR? By melting snow leaving behind the below ice surface?**

Response 2.19: We have incorporated this remark into the text; it sounds interesting as a suggestion.

*"While snow-related clusters nearly disappeared from the northern hemisphere in May 2019, strong LSR signal remained over the Arctic, perhaps indicating a localized peak in sea ice seasonality in the region. Alternatively, there can be an effect of potential wetting of the ice/snow with warmer temperatures behind the increasing LSR because by as snow melts, the below ice surface emerges, potentially contributing to this signal."*

**Line 490: It would be good to have this rough classification and allocation of LSR values to surface types already before Fig. 6.**

Response 2.20: Please note that it is not allocation of colors to certain surface type but the result of Jenks clustering, which yielded from visual point of view highest cluster (red-orange) to snow, mid-clusters to land and low reflectance cluster (blue), mostly to water. See the explanation paragraph about Jenks clustering at pages 22-23.

*"Since Aeolus LSR has been previously shown to reasonably reflect several land cover-related gradients on the map such as: water – land, vegetation – arid, no snow – snow gradients, we used a clustering method to classify the LSR signal for better illustration purposes. To this end, we used natural breaks-based clustering of LSR data for plotting (e.g. Jenks clustering method) for identifying breakpoints between different clusters of LSR data [Sadeghfam et al., 2016]. The method minimizes the average deviation (e.g. variance as well) of each class from its respective mean, concurrently maximizing the divergence of each class from the means characterizing the other classes [Jenks, 1967]. Note that we simply applied this approach for clustering and visualization purposes without intention to disentangle physical differences behind reflectivity patterns of different regions."*

**Line 498/499: Why only the Southern Ocean? I see a lack of valid signals over all ocean areas between +/-70° lat.**

Response 2.21: If you look closely on Fig. 10 and Fig. 11 (a,b), you see some ocean returns from Atlantics and many ocean returns from Pacific Oceans. This is not the case for Southern ocean though with the lowest number of returns among all oceans (visually) which prompted us to make this remark.

**Line 504: I could well imagine here a 4-by-4 plot with maps of the number of valid Aeolus observations per grid point, in order to get an idea of the distribution and significance of the comparison for certain regions.**

Response 2.22: We provide an additional plot with this statistics here for 2019-03 (as example); we have not added such plots in the study because the article contains >15 figures only in the main text and becoming very long to read.

[Figure]

**Line 524: What's the reason for showing the y-axes in log scale, unlike for Fig. 11+12?**

Response 2.23: We replotted scatterplot figures (that were 11 and 12 in previous version) in log-scale as well. See response 1.8 above please to see the reason behind choosing LSR log-plotting.

**Line 814: I would recommend explicitly mentioning the excellent sensitivity of Aeolus LSR to white surfaces (snow/ice conditions) in your abstract.**

Response 2.24: See new remark in the abstract:

*"Regionally, the LSR-LER agreement can vary and yields the highest correlation values in regions where snow is present in winter,* ==*indicating the excellent sensitivity of Aeolus LSR to white surfaces such as snow.*==*"*

**Technical corrections (typing errors, etc.):**

**Line 12: Although well known, the first appearance of the abbreviation "UV" could be written out.**

Response 2.25: Fixed.

**Line 24: Assure consistency for LERG and LERT naming with (l. 137 ff, l. 518 ff., 540 ff., 572, 604 ff.) and without subscripts.**

Response 2.26: Fixed, in long version – LER GOME-2 and LER TROPOMI, in short versions – only LERG and LERT, without subscripts.

**Line 33: The "we demonstrated" seems to be either a leftover or misplaced (parentheses?) to me.**

Response 2.27: This part of abstract was rephrased

**Line 49: … at the lidar …**

Response 2.28: Fixed.

**Line 100: I assume the either "both" or "two" is redundant.**

Response 2.29: Fixed.

**Line 111: As you explain in line 114 that FM stands for feature mask, you might also want to state what PRO stands for.**

Response 2.30: Explanation is given.

**Line 119: described**

Response 2.31: Fixed.

**Line 150/151: No comma needed between "estimates" and "reflected".**

Response 2.32: Fixed

**Line 152/153: Because of the use of "although" this seems to be an unfinished sentence.**

Response 2.33: Fixed.

**Line 176: … where the DEM …**

Response 2.34: Fixed.

**Line 179: … from the ground …**

Response 2.35: Fixed.

**Line 182: Comma after "Aeolus bin" really necessary?**

Response 2.36: Fixed here and elsewhere.

**Line 188: Is "sought" correctly used here? Alternatively: "searched", "looked for" or "sought for" (also line 176).**

Response 2.37: Fixed.

**Line 189: … below the height of the DEM …**

Response 2.38: Fixed.

**Line 189: Proposal: "The algorithm searches for the highest bin that contributes to the ground signal, starting from the first bin with non-negative useful signal."**

Response 2.39: What about this formulation from Response 2.11 where you had already provided some simple suggestion:

*"In short, the Aeolus ground detection looks for signal drops going upwards (in terms of altitude) first and then looks for signal drops going downwards."*

**Line 191: Proposal: "… in upward direction."**

Response 2.40: This is reformulated according to 2.11

**Line 191: lowermost**

Response 2.41: This is reformulated according to 2.11

**Line 199: What shall be expressed by the "-" between "potentially" and "noise"?**

Response 2.42: See rephrased version:

*"However, most cases with no ground bin detected originate from ocean areas with very weak water returns, manifesting a signal of very low magnitude (potentially, it is noise)."*

**Line 222: … and the highest …**

Response 2.43: Fixed.

**Line 226: Eq.1 should most probably start with "LSR'" instead of "y,". Otherwise, introduce y' in the text here, as done with y before Eq.4.**

Response 2.43: Fixed, explanation provided in the text.

---

## Author Response (AR3)

**RESPONSE TO REVIEWERS**

Main author: I apologize for my oversight with numerous technical comments at the previous revision stage. I accidentally cut them from the original response file while merging response to two different reviewers at the first stage of revision. Now, I have noticed this mistake and have incorporated all the responses to these overlooked technical comments in this response (see blue-colored text below), while gray-colored text below indicates the comments that had been already responded at previous stages. Fortunately, most these comments were technical and did not require substantial reworking on our material or overlapped with more major comments on figures or rephrasing that had been placed earlier.

**RESPONSE TO REVIEWER#1**

**1. The methodology in this manuscript should be well-constructed, it should work fine in this study. My main concern is the framework of this manuscript, regarding the sequence of section 3.1 and section 3.2. Normally, one should firstly verify the method proposed in this paper before the statistical analysis. I mean, probably authors may think about describing section 3.2 before section 3.1.**

Response 1.1: We agree with you on this remark. We have now revised the manuscript to present section 3.2 before section 3.1, as you suggested. Additionally, we would like to highlight that the method has already been validated on a limited scale in our previous work (Labzovskii et al., 2023; https://www.nature.com/articles/s41598-023-44525-5), which provides a foundational basis for its more extensive examination in the current study.

**2. For the sections of "Discussion" and "Conclusion," I propose the authors to reorganize these two parts. Some information in the Discussion part should be moved to the "Conclusion" part. In the current manuscript, the conclusion somehow looks like an outlook.**

Response 1.2: Often, researchers read only introduction and conclusions, making this section essential to inform them about key takeaways. In this light, we believe that well-supported conclusions provide an excellent opportunity to offer a perspective of researchers on the scientific findings in either the form of summary or outlook. We opted to frame this perspective as an outlook because the Aeolus mission was the first to apply this methodology to UV LSR. Since Aeolus has been decommissioned we decided to provide in conclusions Aeolus-based guidance and recommendations for future and upcoming satellites. This decision aims to assist researchers interested in implementing such algorithms but who may not be fully aware of the exact opportunities and challenges of highly non-nadir UV LSR. Meanwhile, our discussion summarizes abundant results this study provided in a convenient, short way. I hope the current structure, being supported by such rationale is acceptable.

**Technical corrections:**

**3. The symbols of variables in the article should be consistent. For example, qflag (sometimes quality flag)/LERG (sometimes LERG)/LSR (sometimes lsr)/ODray (sometimes RayOD). Please make them consistent.**

Response 1.3: We standardized all annotations here. Full name TROPOMI LER (when no space – LERT), full name GOME-2 LER (when no space – LERG; full

name quality flag (when no space – qflag); RayOD was a typo. All inconsistencies and subscripts addressed.

**4. Line 317: Why 1-sigma but not 3-sigma is selected? Is 1-sigma too strict for the calculation?**

Response 1.4: The use of 1-sigma is appropriate in this context to ensure that we capture statistically significant variations while maintaining robust results. It is not uncommon to use 1 sigma as the uncertainty proxy for gridded atmospheric datasets in research (see Han et al., for example - https://essd.copernicus.org/articles/15/3147/2023/essd-15-3147-2023.html). Note that this is arbitrary choice that has not affected results in any way.

**5. Figure 3: Red frame: qflag > 0, but the QF in the attenuated case with number 1 is 0. Please check it.**

Response 1.5: The formulation on the figure caption was somehow misleading, despite being technically correct. We rephrased the statement in the middle of Fig. 2 for clarity like *'clear observations: AOD < 1.0; qflag = 0)'*. To clarify here, only the cases with qflag = 0 are considered clear and therefore, only such cases are used in the final analysis.

**6. Figure 4: I cannot see the histogram distributions with magenta color. In the text, of course, you explain only a few numbers of strong return remain; therefore, please find another way to show the distributions of water reflectivity between -35 and 35.**

Response 1.6: We have addressed the issue with Figure 4 and have updated the histogram distributions to improve visibility. The new presentation on Fig. 4 illustrates three distributions with log y-scale, whereas water returns between -35 and 35 latitude are shown in red, rather than transparent magenta, which also improves visual comprehension aspect.

**7. Actually, the authors should reproduce all the figures in this manuscript. They are hard to be recognized, especially for the legends/values.**

Response 1.7: We acknowledge that many figures were initially provided at a resolution lower than dpi 300, while having misalignment issues and caption ambiguities. All figures have been finalized; we have now replaced all figures with higher resolution (dpi 300) to enhance clarity. We have modified most figures to improve their readability, as specified below. For figures with multiple panels, we have made efforts to optimize the quality at 100% zoom. Except dpi enhancement and optimizing the font for 100% zoom at Microsoft Word (as required by the AMT formatting guidelines), we list specific changed below (note that we use the original figure numbering below from version 1 of the manuscript, before sections 3.1 and 3.2 were flipped)

- Figure 1 (all panels and captions aligned)
- Figure 2 (the font of central panel is enlarged; unclarity on the remarks about filtering procedure on '3' is addressed; the statement rephrased to *"Filtering out low-quality LSR (clear observations: AOD < 1.0; QFLAG = 0)"*)
- Figure 3 (only dpi and fonts changed)

- Figure 4 (For better visibility, we changed y-axis to log-scale visualization and changed magenta transparent color for denoting no-ice water to red non-transparent color; other changes are cosmetic and were applied to improve the aesthetics of the figure)

- Figure 5 (observation points increased, panel annotations 'abcd' assigned, captions aligned, frames created)

- Figures 6 – 8 (replotted with higher dpi, panels were annotated 'abcd…' and aligned)

- Figure 10 (the size is significantly enlarged; every panel was annotated as a, b, c, d)

- Figures 11 – 13; 16 – 18 (Panels annotated as 'abcd…', panels aligned, titles added, captions corrected; z-axis color annotated)

- Figure 14 (panels annotated, region annotations aligned, y-axis changed to "LER" to avoid confusion)

- Figure 15 (the font size of correlation caption increased, the figure size increased as well)

**8. Figure 10: Is it necessary to use log scale for LSR?**

Response 1.8: The use of a logarithmic scale for LSR is advisable due to the exponential nature of the LSR signal, which changes significantly from water to snow surfaces. Its behavior is in this aspect different from LER that changes more linearly and uniformly as the surface becoming whiter. In sort, the logarithmic representation effectively captures the wide range of signal variations. We have documented this phenomenon in our previous study (Labzovskii et al., 2023).

**9. Figure 13: Why only "LER snow TROPOMI" and "LER CLEAR GOME-2" are compared? The authors should also explain why LER with other features are not compared.**

Response 1.9: This rationale stems from our previous study (Labzovskii et al., 2023; https://doi.org/10.1038/s41598-023-44525-5) and was previously described. To quote the Data section from Labzovskii et al. 2023: *We used LER estimates, reflecting snow-affected areas as well because it is crucial to include snow/ice regions in the analysis to evaluate the sensitivity of Aeolus LSR to the strongest white reflectors*. Note that for GOME-2 this is the only available type of dataset at 354 nm we compared with. A more detailed explanation is given in 2.1.2 of the current manuscript.

**10. Figure 14: Which y-label is designed for TROPOMI? Is the y-label of "GOME-2 LSR" shared for TROPOMI and GOME-2? Please make it clear.**

Response 1.10: The caption here was misleading. Y-axis reflects LER for both GOME-2 and TROPOMI (despite being not identical, they are very similar in average terms and let alone, similar by magnitude). We have corrected the y-axis to 'LER' in this Figure 14.

**11. Line 680: How the values of snow cover (SNW_1) are calculated? Is it a proportion?**

Response 1.11: Snow cover information is taken from MERRA-2; an atmospheric reanalysis system from the NASA GMAO efforts (Gelaro et al. https://journals.ametsoc.org/view/journals/clim/30/14/jcli-d-16-0758.1.xml).

MERRA-2 uses the GEOS-5 model to simulate atmospheric processes like radiation and convection on a grid. It employs Gridpoint Statistical Interpolation (GSI) to integrate observational data, adjusting the model's outputs to reflect real-world conditions every six hours. In terms of sea ice, MERRA-2 assimilates recalibrated versions of some of the satellite observation types like for sea ice observations, primarily obtained from satellite passive microwave sensors, provide data on sea ice concentration and extent. We added this rephrased remark from GEOVANNI website description of snow/ice cover extent: "*...where the extent of snow or ice cover on the Earth's surface is used (the fractional amount of a land surface covered with snow and ice within a tile, ranging from 0 to 1).*"

**12. Figure 17: Please recheck this figure. The titles (month information) of the graph in the first row have been obscured and are not displayed entirely.**

Response 1.12: We have revised all figures, see Response 1.7 please.

**13. Line 738: Discussion should be listed as section 4.**

Response 1.13: Discussion is now labelled as section 4.

**14. Line 809: Conclusions should be listed as section 5.**

Response 1.14: Conclusions are now labelled as Section 5.

**RESPONSE TO REVIEWER #2**

**Specific comments (addressing individual scientific questions/issues):**

**Line 1: Why is "Phase-F" mentioned so prominently in the paper title? Isn't "new" enough information? As "Phase-F" appears only one more time in the whole manuscript, I recommend deleting this term from the title.**

Response 2.1: We deleted this from the title.

**Line 10: The term "incidence angle" might need additional reference information (incident onto what?). Maybe consider using the term "off-nadir" angle. (Further cases in lines 67, 97, …)**

Response 2.2: We provided the definition of incidence angle at the first occurrence in the introduction, to quote the introduction "*incidence angle of satellite instrument is the angle between the satellite sensor and the normal to the surface of the target cell*." We kept the incidence angle in the abstract without explanation which is not uncommon for lidar studies. Namely, Kassalainen et al., 2018 is one among many examples we can provide - https://royalsocietypublishing.org/doi/full/10.1098/rsfs.2017.0033.

**Line 22: Why are the reference orbits only from the period where Aeolus used its laser A? What about data from laser B with L1B 7.12? Please mention this including the used baseline version. See also line 107 and Table 1. Are different results or performances expected from FM-B data?**

Response 2.3: Regarding the baseline version, we used the #3 reprocessing results in baseline 14 for FM-A period. We incorporated this remark in the methodological description where you pointed out. We used only FM-A period in this paper, from the reprocessing perspective, we have data input dependence with not only Aeolus L1 and L2A data, but also AEL-PRO algorithm alongside AEL-FM mask within. At the moment of the manuscript submission, we could rely only on the input from the FM-A period (before June 2019) in data

dependency. For the reprocessing #2 (FM-B) baseline 11 was used. These products did not contain the necessary variables to run the AEL-PRO (prototype) algorithm. Reprocessing #4 (FM-B part) was not yet released during this work so could not be used. Moreover, thus far, we have officially produced LSR datasets only for FM-A period using the #3 reprocessing and baseline 14 only, available for open access sharing but the FM-B period datasets should be produced soon from our side and we'll analyze this aspect independently. Some differences between FM-A and FM-B LSR outputs will be plausible according to our previous experience.

**Line 36: It might be worth noting shortly here that Calipso measured at a different wavelength than Aeolus.**

Response 2.4: We included this remark.

*"By taking together* ==nadir-looking CALIPSO and highly non-nadir Aeolus experiences,== *a framework on effective LSR utilization using future lidar missions such as EarthCARE and Aeolus-2 can be effectively designed."*

**Line 67: What do you refer to here specifically with "unique Aeolus setup"? In which way does it constitute a challenge to retrieve robust LSR? Presumably you mean the coarse vertical resolution.**

Response 2.5: We literally meant the highly-non nadir incidence of ~35° and 355 nm wavelength. This combination is unique itself for atmospheric lidar research regardless vertical resolution or other characteristics that may differ from CALIPSO for example.

**Line 120: Table 1. As you introduce the "reprocessing product" in line 107, it would be good if you add the Baseline to the column "Version" at least for the L1B.**

Response 2.6: The remark on the reprocessing # and baseline added in the paragraph over Table 1.

**Line 139: Could you provide a reference that shows how real surfaces can behave in terms of angular dependent reflection?**

Response 2.7: We added a reference from Maignan et al., (https://www.sciencedirect.com/science/article/pii/S0034425703003808?casa_token=cdF2rk0M74YAAAAA:JK-mZi4v_9D7788JC6KoHfj61sy0Hm0-2noc7fQFxkemMCn8TMFoIq_Rfr6Ru946M4moD_zC) study, who elucidated, to quote how to …"*to measure the bidirectional reflectance of a large variety of Earth targets*". For 22 000 sets of measured targets and evaluated reflectance models using real observations.

**Line 149: Why do you perform a resampling from 0.25° to a coarser resolution of 2.5°?**

Response 2.8: Because at 0.25°, there is an insufficient amount of Aeolus data to fill the grid cell. Therefore, grid comparison between LSR and LER should be done at a realistic resolution like 2.5°. This decision was made based on our previous experience from Labzovskii et al. (2023) that set our expectations for the balance between data abundance and the realistic resolution, of potential LSR L3 gridded product.

**Line 165: Why can't NDVI become negative (see equation in line 164)? Is the absolute value of NIR always equal to or larger than that of VIS?**

Response 2.9: In short, yes, for most vegetated areas, it cannot be negative because healthy vegetation reflects more NIR light than red light, making the difference between NIR and Red positive. Negative NDVI can occur over water or urban areas for example; both irrelevant to our study. We added a remark on this

*"Negative NDVI values occur in scenarios where the reflectance properties are not typical of vegetation, like water, but such areas are outside of the scope of our analysis."*

**Line 187: To my knowledge, the Aeolus mission uses the ACE-2 DEM: https://sedac.ciesin.columbia.edu/data/set/dedc-ace-v2.**

Response 2.10: We corrected the description to ACE v.2 DEM.

**Line 189 ff.: To my knowledge, the Aeolus ground detection looks for signal drops going upwards (in terms of altitude) first and then looks for signal drops going downwards.**

Response 2.11: Perhaps, our description was too vague. Please see if our current description incorporating your simpler, plainer explanation is better

*"The height of the surface of the Earth with regard to the reference ellipsoid is used. Subsequently, the lower edge of each altitude bin is being sought, where the height of the bin should be below height of DEM. In short, the Aeolus ground detection looks for signal drops going upwards (in terms of altitude) first and then looks for signal drops going downwards. If ground bin candidates are more*

*than five, the ground detection is not successful and therefore no ground bin is assigned for the respective observations [Lux et al., 2018]."*

**Line 200: Northern or southern hemisphere winter in general, or both? Or do you refer to cases b and c?**

Response 2.12: We rephrased that we did refer to the northern hemisphere here.

*"According to our previous experience [Labzovskii et al. 2023], more detected cases in winter in northern hemisphere are explained by the presence of sea ice over ocean. As we are interested in land surface signal, we applied the surface flag mask"*

**Line 226: Assuming that the ground signal (disregarding the atmospheric contribution to the signal of the bin) emanates from a more or less infinitesimally thin layer, why shall it make sense to multiply here with the thickness of the whole bins (apart from getting a useful unit)? Doesn't the variation of the range-bin-thickness (0.25 km – 2 km, line 97) inadequately affect the LSR'?**

Response 2.13: Apologies for uploading an empty response to this comment, I've accidentally uploaded the older version without this comment being implemented The problem is that the units and names of variables were mixed up in the previous version. We updated the description here by adding units for clarity, see the inclusion as :

*In calculating LSR, we used all ground bin numbers marked as containing surface (see 'ground_bin_num' in Aeolus L1A data). As in Labzovskii et al. [2023], we took the attenuated backscatter (β, $sr^{-1} m^{-1}$) at range (z) from the AEL-PRO L2 data at the bins where ground was detected and multiplied it to the width of the surface range bin of Aeolus ($\Delta r_{surf}$, m). In this way, we obtained the uncorrected Surface Integrated Attenuated Backscatter (SIAB', $sr^{-1}$) or, in terms of this paper – the uncorrected Lidar Surface Returns (LSR', $sr^{-1}$), reflected as (γ') in Eq. 1. The ground location was determined using the lowest bin where the ground is located ($s_{min}$, m) and the highest bin where ground is located ($s_{max}$, m)*

**Line 365: Can sea ice already be expected close to these latitude thresholds, e.g., around +/-40°?**

Response 2.14: The result here can vary depending on the methodology applied and on the assumption. For your interest, we duplicated the figure with -40 to +40 latitude boundaries below. Since only miniscule differences between two choices and respective distributions are seen, let us keep the original figure with -35 < lat < 35 boundaries.

[Figure]

**Line 367: What should be the reason for a low amount of sea ice right after the southern hemisphere winter?**

Response 2.15: There are some studies we examined. To name one, Holland, (2014;

https://agupubs.onlinelibrary.wiley.com/doi/epdf/10.1002/2014GL060172)

showed that *"the ice-albedo feedback causes instability during **spring and summer**, whereby ice cover perturbations grow"* (e.g. spring ice loss is heavy due to perturbations). Other reasons are wind trends and temperature effects (reduction trend starts exactly after winter in southern hemisphere near Antarctica).

**Line 429: Do you have any idea what could be the reason for the ocean returns being placed exclusively in such regions? Algae (might explain the seasonality), white caps (maybe rather the cyan values around -60° latitude?), microplastics, …? It might be worth putting the info from line 474/476 already here?**

Response 2.16: It is a very difficult pattern to disentangle because the official Aeolus algorithm filters out what is labeled as weak signal. However, we have a working hypothesis based on the letter we are preparing on ocean returns (mentioned in response 2.14), where we evaluated also assumingly weak ocean surface returns. It is a complex interplay between chlorophyll concentration and ocean surface conditions. In short, Aeolus ocean LSR is strong if chlorophyll is low and ocean mixing is low (tropical and sub-tropical gyres) with ~0.2 mg/m³ of chlorophyll as a threshold. Otherwise, once chlorophyll grows above this threshold, complete attenuation of Aeolus UV signal occurs and the signal becomes too weak or statistically disappears. We are trying to prove the validity of this pattern in the aforementioned letter and, unfortunately, cannot focus on this aspect more or provide this hypothesis without additional analysis and figures in the current paper.

**Line 440: Figure 6, top left panel: Could you comment on the apparent increase of valid data in Africa (Sahel zone, region south of African rainforest, Mozambique), South America (north and south of the rainforest), and South East Asia from September to November? Is it related to the general unavailability of data, or could it have a meteorological (or another) reason, such as the movement of the ITCZ or cloud climatology? Or are lines 453-457 sufficient for explanation?**

Response 2.17: It can be related to ITCZ and also to the seasonality of west Africa biomass burning. We internally discussed a lot on this phenomenon. However, we do not have numerical arguments to prove this except for the actual result of these phenomena—namely high AOD from Aeolus. Due to this, we restricted ourselves to the explanation in lines 453-457 that is based on numerical AOD findings we have. In other words, we do not do deeper into the explanations we cannot prove here.

**Line 460: Did you encounter a decrease in the number of valid LSR data over the FM-A period correlated with the decrease in the laser energy of Aeolus?**

Response 2.18: We suspected this in terms of correlation and abundance, but it was not the case. See below the statistics of the successful ground detections (GD, official algorithm), cloud-free detections with aerosols (second column) and clear LSR estimates for the final analysis (third column). The abundance is more driven by cloudiness around the world, while for the global reflectivity, starting from January, global signal starts decreasing towards September (absolute minima), so downward trend towards May is plausible for this kind of data.

| date | All | Ground Detected (GD) | GD & qflag = 0 | GD & qflag = 0 & AOD < 1 |
|---|---|---|---|---|
| 2018/09 | 5,480,310 | 465,044 | 393,218 | 366,529 |
| 2018/10 | 6,398,400 | 645,696 | 520,056 | 482,108 |
| 2018/11 | 5,937,690 | 1,115,751 | 874,464 | 808,765 |
| 2018/12 | 6,213,600 | 1,255,028 | 972,504 | 898,912 |
| 2019/01 | 2,760,390 | 605,257 | 470,673 | 436,531 |
| 2019/02 | 2,659,800 | 520,681 | 359,191 | 330,899 |
| 2019/03 | 6,512,100 | 1,324,018 | 1,128,701 | 1,026,869 |
| 2019/04 | 6,263,430 | 1,168,668 | 949,454 | 859,650 |

| 2019/05 | 6,507,330 | 959,369 | 728,891 | 660,989 |
|---|---|---|---|---|

**Line 466/467: To me, this doesn't look localized, but rather like a clear trend for the whole Arctic region for these three months, as also Greenland shows this behavior. But can a potential "wetting" of the ice/snow with warmer temperatures be consistent with increasing LSR? By melting snow leaving behind the below ice surface?**

Response 2.19: We have incorporated this remark into the text; it sounds interesting as a suggestion.

*"While snow-related clusters nearly disappeared from the northern hemisphere in May 2019, strong LSR signal remained over the Arctic, perhaps indicating a localized peak in sea ice seasonality in the region. Alternatively, there can be an effect of potential wetting of the ice/snow with warmer temperatures behind the increasing LSR because by as snow melts, the below ice surface emerges, potentially contributing to this signal."*

**Line 490: It would be good to have this rough classification and allocation of LSR values to surface types already before Fig. 6.**

Response 2.20: Please note that it is not allocation of colors to certain surface type but the result of Jenks clustering, which yielded from visual point of view highest cluster (red-orange) to snow, mid-clusters to land and low reflectance

cluster (blue), mostly to water. See the explanation paragraph about Jenks clustering at pages 22-23.

*"Since Aeolus LSR has been previously shown to reasonably reflect several land cover-related gradients on the map such as: water – land, vegetation – arid, no snow – snow gradients, we used a clustering method to classify the LSR signal for better illustration purposes. To this end, we used natural breaks-based clustering of LSR data for plotting (e.g. Jenks clustering method) for identifying breakpoints between different clusters of LSR data [Sadeghfam et al., 2016]. The method minimizes the average deviation (e.g. variance as well) of each class from its respective mean, concurrently maximizing the divergence of each class from the means characterizing the other classes [Jenks, 1967]. Note that we simply applied this approach for clustering and visualization purposes without intention to disentangle physical differences behind reflectivity patterns of different regions."*

**Line 498/499: Why only the Southern Ocean? I see a lack of valid signals over all ocean areas between +/-70° lat.**

Response 2.21: If you look closely on Fig. 10 and Fig. 11 (a,b), you see some ocean returns from Atlantics and many ocean returns from Pacific Oceans. This is not the case for Southern ocean though with the lowest number of returns among all oceans (visually) which prompted us to make this remark.

**Line 504: I could well imagine here a 4-by-4 plot with maps of the number of valid Aeolus observations per grid point, in order to get an idea of the distribution and significance of the comparison for certain regions.**

Response 2.22: We provide an additional plot with this statistics here for 2019-03 (as example); we have not added such plots in the study because the article contains >15 figures only in the main text and becoming very long to read.

[Figure]

**COUNT OF OBSERVATIONS FOR GRIDDING (2019 – 03 ) WITH QFLAG = 100% (CLEAR OBSERVATIONS)**

**Line 524: What's the reason for showing the y-axes in log scale, unlike for Fig. 11+12?**

Response 2.23: We replotted scatterplot figures (that were 11 and 12 in previous version) in log-scale as well. See response 1.8 above please to see the reason behind choosing LSR log-plotting.

**Line 814: I would recommend explicitly mentioning the excellent sensitivity of Aeolus LSR to white surfaces (snow/ice conditions) in your abstract.**

Response 2.24: See new remark in the abstract:

*"Regionally, the LSR-LER agreement can vary and yields the highest correlation values in regions where snow is present in winter, indicating the excellent sensitivity of Aeolus LSR to white surfaces such as snow."*

**Technical corrections (typing errors, etc.):**

**Line 12: Although well known, the first appearance of the abbreviation "UV" could be written out.**

Response 2.25: Fixed.

**Line 24: Assure consistency for LERG and LERT naming with (l. 137 ff, l. 518 ff., 540 ff., 572, 604 ff.) and without subscripts.**

Response 2.26: Fixed, in long version – LER GOME-2 and LER TROPOMI, in short versions – only LERG and LERT, without subscripts.

**Line 33: The "we demonstrated" seems to be either a leftover or misplaced (parentheses?) to me.**

Response 2.27: This part of abstract was rephrased

**Line 49: … at the lidar …**

Response 2.28: Fixed.

**Line 100: I assume the either "both" or "two" is redundant.**

Response 2.29: Fixed.

**Line 111: As you explain in line 114 that FM stands for feature mask, you might also want to state what PRO stands for.**

Response 2.30: Explanation is given.

**Line 119: described**

Response 2.31: Fixed.

**Line 150/151: No comma needed between "estimates" and "reflected".**

Response 2.32: Fixed

**Line 152/153: Because of the use of "although" this seems to be an unfinished sentence.**

Response 2.33: Fixed.

**Line 176: … where the DEM …**

Response 2.34: Fixed.

**Line 179: … from the ground …**

Response 2.35: Fixed.

**Line 182: Comma after "Aeolus bin" really necessary?**

Response 2.36: Fixed here and elsewhere.

**Line 188: Is "sought" correctly used here? Alternatively: "searched", "looked for" or "sought for" (also line 176).**

Response 2.37: Fixed.

**Line 189: … below the height of the DEM …**

Response 2.38: Fixed.

**Line 189: Proposal: "The algorithm searches for the highest bin that contributes to the ground signal, starting from the first bin with non-negative useful signal."**

Response 2.39: What about this formulation from Response 2.11 where you had already provided some simple suggestion:

*"In short, the Aeolus ground detection looks for signal drops going upwards (in terms of altitude) first and then looks for signal drops going downwards."*

**Line 191: Proposal: "… in upward direction."**

Response 2.40: This is reformulated according to 2.11

**Line 191: lowermost**

Response 2.41: This is reformulated according to 2.11

**Line 199: What shall be expressed by the "-" between "potentially" and "noise"?**

Response 2.42: See rephrased version:

*"However, most cases with no ground bin detected originate from ocean areas with very weak water returns, manifesting a signal of very low magnitude (potentially, it is noise)."*

**Line 222: … and the highest …**

Response 2.43: Fixed.

**Line 226: Eq.1 should most probably start with "LSR'" instead of "y,". Otherwise, introduce y' in the text here, as done with y before Eq.4.**

Response 2.43: Fixed, explanation provided in the text.

**21.Line 226: Eq.1 – missing explanation of zi**

Response 2.44: We fixed this, see response 2.13 please.

**22.Line 244: … the optical depth we …**

Response 2.45: This has been rephrased as a result of the other comment implementation.

**23.Line 250: overestimation**

**24.Line 250: probably two spaces after "situated"**

**25.Line 255: … the given ??? …**

**26.Line 263: aerosol**

**27.Line 266: two commata after "(HSRL)"**

**28.Line 270: RayOD □ ODRay**

**29.Line 271: deleted the second "both" in the same sentence**

**30.Line 280: … of the LSR …**

Response 2.46: All the typos above corrected

**31.Line 281: Please explain/write out AB already here than in line 329.**

Response 2.47: AB term is now explained in the segment of article, where you point at.

**32.Line 295: probably two spaces after "echo"**

Response 2.48: All unnecessary double-spacing cases are amended here and elsewhere.

**33.Line 318: … calculating the scientific …**

**34. Line 320: probably two spaces after "year"**

**35. Line 324: Fig.2 contains "L1A" in the top box, although this has not yet been mentioned. Perhaps you mean L1B.**

Response 2.49: It was a serious typo and we corrected it in the current version (L1B and AEL_PRO data is mentioned now)

**36. Line 330: orbits**

Response 2.50: Typo corrected

**37. Line 330: Proposal: "… with colored indexed for the selected observations placed in each subplot."**

Response 2.51: I did not understand what is "colored indexed" in this context. I rephrased this part as: *"Below, we show some examples with LSR vertical profiles from the reference orbits (Fig. 3). We plotted vertical profiles of ==attenuated backscattering== from Aeolus observations with markers, signifying the ground detection (==see red circles on the vertical profiles, placed== on Fig. 3). All these cases are taken from the reference ==orbits== we described before with the index of observational point expressed over top of each subplot."*

**38. Line 334: … or weakened the LSR …**

Response 2.52: Initial formulation is correct, we did mean "weaker than LSR"

**39. Line 336: qflag**

**40. Line 340: … having the highest …**

**41. Line 348: … ground bins …**

Response 2.53: All typos above corrected

**42. Line 352: Fig.3 could have a more expressive design, e.g. with filled, colored and semi- transparent background boxes.**

Response 2.54: It could have different design solutions, but I preferred a minimalistic design with non-lush colors here due to excessive amount of elements of this figure. It is easier to read such complex figures with rather functional than expressive design from my point of view.

**43. Line 353: (ATB) should be deleted here, because it defines a second acronym for a quantity that has already been defined before.**

Response 2.55: Done!

**44. Line 354: According to the text above case 1 (red frame) has a qflag = 0%.**

Response 2.56: A typo here; two conditions here: qflag > 0% or AOD > 1.0

**45. Line 357: Proposal: "Note that the term index above each subplot stands for the respective orbit number of the Aeolus mission."**

Response 2.57: Correction applied

**46. Line 363: This is not a full sentence.**

Response 2.57: Correction applied:

*It reflects the distribution of LSR for every reference orbit selected earlier for three types of surfaces: land, water and water in low latitude regions (black, blue and magenta colors in Fig. 4, respectively).*

**47. Line 383: Would be nice to have all four plots with the same x-axis range.**

Response 2.58: See new version of the figure with x-axis range from 0 to 1.

**48. Line 384: latitude**

Response 2.59: Correction applied

**49.Line 370/371: There is no error propagation mentioned in the caption of Figure 4.**

Response 2.60: We added a remark for a reader that we mention errors only in the text, not on this plot (these are distribution plots):

*Note that these errors are not reflected on Fig. 4, where only distributions are shown.*

**50.Line 374: Within the parentheses the values for water seem to be missing.**

Response 2.61: In the new version, we removed a remark about land and water errors because it was rather confusing in the V1 form for a reader.

**51.Line 374: … land are explained …**

Response 2.62: This part does not exist in the current version.

**52.Line 377: … are clipped for water surface from …**

**53.Line 388: covers**

**54.Line 392: probably two spaces after "hypotheses" and before "strong"**

**55.Line 398: probably two spaces after "we"**

**56.Line 399: probably two spaces after "Several"**

Response 2.63: All corrections from above applied; all typos corrected.

**57.Line 440: The caption mentions panels a-f, but the letters do not appear anywhere.**

Response 2.64. Fig. 5 as well as other figures were replotted. This Fig. 5 now has a, b, c, d panels.

**58.Line 429 (same for line 460+477+503): The title textboxes of the lower four plot seem to overlap with their above plots. Resolution of the figure could be improved. What about the information on the top left? Lat/Lon info for the y/x axes would be helpful.**

Response 2.65. Fig. 5 was replotted, as well as other figures. As mentioned in one of the responses above, the following corrections applied: observation points increased, panel annotations 'abcd' assigned, captions aligned, frames created. Let us know if this form of figure is good.

**59.Line 490: Proposal: … representing mostly sea ice … (red colour appears also elsewhere)**

Response 2.66: Suggestion applied

**60.Line 533: Is this a second caption to the same table? Please unify the information.**

Response 2.67: Second caption has been removed.

**61.Line 540: LER instead of LERs**

Response 2.68: Suggestion applied

**62.Line 542 ff.: Please insert the values for the correlation coefficients also in Figures 11 and 12, as done for Fig. 13.**

Response 2.69: We mentioned correlation coefficients in the text above instead of figures not to overburden the plots visually, you can see some plots like 2018-09 are completely covered by scattering points.

**63.Line 547: … along the y-axis …**

**64.Line 548: Proposal: "various snow types and conditions" instead of just "snow"**

**65.Line 548: … by a strongly …**

**66.Line 553: … such an experimental …**

**67.Line 556: … is increased to r = …**

Response 2.68: All suggestions above applied

**68.Line 564: Fig. 11: Not a good resolution. Labels of y-axes overlap with colour scale of neighboured plots. Caption: LER should be LERG or LERG or LERG. Please make consistent with x-axis labels.**

Response 2.69. As mentioned, all figures replotted with high resolution and aligned well including Fig. 6 and Fig. 7 in this version. I did not understand the comment about LERG, LERG and LERG, I assume it was about specifying type of LER we use on the axis. Now, Fig. 6 states: LERG (thus, GOME-2) and Fig. 7 states LERT (thus, TROPOMI).

**69.Line 573: probably two spaces after "the"**

Response 2.70: Suggestion from above applied

**70.Line 575: delete "comparisons"**

Response 2.71: We kept the word "comparison" here because it is clearer for readers what we mean by Aeolus-TROPOMI and Aeolus-GOME-2.

**71.Line 576: Does a second "agreement" make sense here?**

Response 2.72: I think readers can easily make a conclusion about the magnitude of r improvement here without further remarks because we mention numbers

from both works (r = 0.89 from Labzovskii et al. 2023 compared to 0.92 in this work for LERT-LSR and 0.62 to 0.90 for LERG-LSR comparisons).

**72. Line 579: Fig. 12: Not a good resolution. Caption: LER should be LERT or LERT or LERT. Please make consistent with x-axis labels.**

Response 2.73. See response 2.69 please, the same applies to this figure.

**73. Line 586: Fig. 13: Panel a and b are mentioned in the caption but are not noted anywhere in the figure. ☐ use left and right?**

Response 2.74. Panels a and b are now assigned to Fig. 8 (it used to be Fig. 13 in 1st version to clarify for the editor).

**74. Line 598: … from an ecosystem …**

**75. Line 599: … and a correlation …**

**76. Line 601: probably two spaces after "was"**

**77. Line 605: Fig 14c refers to the Sahara region, not to snow covered Scandinavia (a).**

Response 2.75. All typos corrected from above.

**78. Line 607: Where do you see the ~0.3 of LER(G/T?) in Fig. 14a?**

Response 2.76. You need to look at right y-axis (black axis). You may find even slightly above 0.3 values in terms of LER.

**79. Line 608: "Northern Canada" … not shown here**

Response 2.77. We added the following remark for readers (we think it is important to note that not all snow-prone regions follow the pattern we talked about). Indeed, there is no possibility to include all regions mentioned in the

correlation figure later on the detailed regional figure here. *This one-peak curve is evident for both the LER and LSR estimates except in Northern Canada, where LSR decreases at a faster rate in comparison to LER estimates at the end of winter (not shown on the Fig. 14a with different regions).*

**80. Line 610: Where are these values for Central Greenland in Fig. 15?** □

Response 2.78. We created this figure in the way that if we have low correlations, the color is reset to white like in the case of Greenland. Yellow-to-red gradient is switched on only for >0.50 correlation cases. Thus, correlation for Greenland is low (r < 0.50 in both LERG-LSR and LERT-LSR). I suspect that you are asking because the high correlation for Greenland does not agree with white color, indicating no correlation on this figure. But comment 2.79 below explains this point further; there was a typo in the text.

**81. Line 611ff: Do these values match the colours in Fig. 15, or ist the colour scale wrong?**

Response 2.79. Yes, same correlation coefficients for regional analysis! Thanks for noticing the typo because Antarctica and Central Greenland were wrongly flipped here in the text.

**82. Line 614: probably two spaces after "changes"**

**83. Line 619/620: Grammatically incorrect first part of the sentence.**

Response 2.80. Typos from above have been corrected; the grammar corrected.

**84. Line 624: very god agreement only too LERT but obviously not to LERG**

Response 2.81: We put as following *...very good agreement between LSR and LERT over Sahara...*

**85. Line 655: Fig. 14. You show LERT and LERG, but mention only LERG in the labels of the y-axes. What about the availability of error bars for LERT?**

Response 2.82: We replotted this figure with higher resolution, rephrasing LERG to LER on y-axis. Note that both LERG and LERT change within the same range of value, so the same y-axis (right y-axis) is enough. Errorbars of GOME-2 are similar to TROPOMI LER and are not shown for better visibility purposes otherwise figures like Fig. 14b are very badly readable.

**86. Line 625: Fig. 14a refers to Scandinavia but not to Sahara**

**87. Line 625: sensitive ☐ sensitivity**

**88. Line 634: Fig. 14a refers to Scandinavia but not to Sahara**

**89. Line 638: … of the southern …**

**90. Line 643: 15 d ☐ 14 d**

**91. Line 648: Refer to Figure 14 e.**

**92. Line 650: 15 f ☐ 14 f**

**93. Line 714: Fig. 16 caption: … during the entire …**

Response 2.83. All typos and oversights from above are corrected.

**94. Line 714: Fig. 17 is clipped at the top. Caption: … during the entire … colorbar …**

Response 2.84. These figures are completely replotted, no clipped features and misalignment anymore.

**95. Line 695: This is not a fully sentence in a grammatical sense. Also "moreover" doesn't make sense here as you already referred to the pattern in the previous sentence.**

Response 2.85. We rephrased it to: *...During these months, two distinct populations of LSR with negative association were discerned: stronger LSR (see the horizontally prolonged upper population) and the lower LSR population of the same shape...*

**96. Line 699: Fig 16 – 17 – 18: Bad resolution ☐ hardly readable label of y-axes. Please use capital letters for "LSR" in the labels.**

Response 2.86. See response 2.84 please.

**97. Line 703: Please explain the value 0.05 here. What does it mean and where do you get it from? Is it 5% of the area covered by snow?**

Response 2.87. To quote our methodology: *...snow cover data were taken from the MERRA-2 (Modern-Era Retrospective analysis for Research and Applications version 2) model, namely from the M2TMNXGLC dataset, where the extent of snow or ice cover on the Earth's surface is used (the fractional amount of a land surface covered with snow and ice within a tile, ranging from 0 to 1).* In fact, yes, it means 5% in 0 – 100% range. In practice, in this modelled dataset, this is a quantitative threshold below which only the tiles with insignificant amount of snow are present in the modelling output.

**98. Line 705: Fig. E4 ☐ Fig. S8**

**99. Line 706: No need to refer to the same figure again.**

Response 2.88. Both oversights are tackled here.

**100.** **Line 734: Fig. 18. Not well arranged with partly mutual clipping of the subplots. Does "snow cover cases" represent a count/number here? If so, it needs a second y-axis. In Figure 16 it is "snow cover" only. Otherwise formulate the sentence more clearly. Think of a more appropriate label for the x-axes or add a second one for SNW. Are the grey bars error bars?**

Response 2.89. The explanation of this figure suffered from unclarity. First, we replotted the figure at higher resolution and more explicit descriptions. Second, there is no need to plot second axis for snow cover because both NDVI and snow cover change from 0 to 1 and in simple words reflect the amount of snow per tile and amount of green healthy vegetation per tile. Third, we added a new figure description

*Fig. 18 Bin-based plots for LSR averages depending on NDVI with snow (VEG-Y), NDVI without snow (VEG-N) and snow covered areas (SNW) for every month in the FM-A period. Note that every marker in these plots represent LSR average for a certain range of VEG-Y, VEG-N or SNW. We considered 50 bins, which means there are 50 LSR averages, evenly distributed across x-axis from 0 to 1. We illustrate vertical LSR errorbars of VEG-Y and VEG-N, but not SNW for better illustration purposes. Quantitatively, LSR errorbars of SNW are very similar to LSR errorbars of VEG-Y and therefore can be omitted. X-axis represent both NDVI and SNW change from 0 to 1.*

**101.** **Line 727: … in our previous paper …**

**102.** **Line 729: level**

**103.      Line 731: … in the discussion chapter in …**

Response 2.90: All the typos noticed above are corrected.

**104.      Line 740: What do you mean with "during post-commissioning phase"? Doesn't the commissioning phase of a satellite comprise only the first few weeks to months? If so, this would be long ago for Aeolus, but you seemingly want to express that your LSR product is to be placed into the Level 2A product in the near future, which is already after the end of the Aeolus mission.**

Response 2.91: We removed this unnecessary remark.

**105.      Line 751: … in the FM-A …**

**106.      Line 760: … different magnitudes of return signal … effects in the returns …**

**107.      Line 768: Decide for one consistent representation of the ranges, either "x – y" or "x**

**< LSR < y"**

**108.      Line 791: … to a low …**

Response 2.92: All these typos from above have been corrected. Your recommendations were taken into account.

**109.      SUPPLEMENT Line 19: What does "grind" mean in the x-axis labels?**

Response 2.93: This was a typo, we meant "Gridded uncorrected values". See new version of the figure.